# Decomposing Novel into Known: Part Concept Learning For 3D Novel Class Discovery

**Tingyu Weng**      **Jun Xiao**      **Haiyong Jiang**[*]
School of AI
University of Chinese Academy of Sciences
wengtingyu18@mails.ucas.ac.cn,{xiaojun,haiyong.jiang}@ucas.ac.cn

## Abstract

In this work, we address 3D novel class discovery (NCD) that discovers novel classes from an unlabeled dataset by leveraging the knowledge of disjoint known classes. The key challenge of 3D NCD is that learned features by known class recognition are heavily biased and hinder generalization to novel classes. Since geometric parts are more generalizable across different classes, we propose to **d**ecompose **n**ovel **i**nto **k**nown parts, coined *DNIK*, to mitigate the above problems. *DNIK* learns a part concept bank encoding rich part geometric patterns from known classes so that novel 3D shapes can be represented as part concept compositions to facilitate cross-category generalization. Moreover, we formulate three constraints on part concepts to ensure diverse part concepts without collapsing. A part relation encoding module (PRE) is also developed to leverage part-wise spatial relations for better recognition. We construct three 3D NCD tasks for evaluation and extensive experiments show that our method achieves significantly superior results than SOTA baselines (+11.7%, +14.1%, and +16.3% improvements on average for three tasks, respectively). Code and data will be released.

## 1   Introduction

3D recognition is fundamental for 3D tasks and plays a critical role in many applications such as robotics and autonomous vehicles. Recently, deep learning-based 3D recognition [24, 25, 45] has achieved remarkable progress in this task. Unfortunately, most methods are limited to annotated classes in the training set (i.e., known classes), and cannot recognize novel classes like human beings, which limits the robustness of 3D recognition in the real world. Therefore, it is crucial and valuable to investigate the *3D novel class discovery task (NCD)* so that 3D recognition can discover novel classes by leveraging the knowledge of known classes.

Though 2D NCD has been extensively investigated, a straightforward extension of 2D NCD methods [7, 19] to 3D NCD usually performs poorly due to the *feature bias problem* [32], *i.e.*, the learned features are heavily biased towards known classes and have difficulty in generalizing to novel classes. The main factors lie in two aspects. First, learned feature representations are usually dominated by known classes while being oblivious to novel classes. Second, 3D recognition usually focuses on shortcut features [8, 12] only capture a special global structure or local regions specific to known classes [21] as shown in Fig. 1.

In this work, we tackle the above problems based on the observation that humans can subconsciously decompose shapes into familiar part concepts, which along with their spatial relations provide effective cues for recognizing different novel shapes [13, 18]. Therefore part concepts can bridge the gaps between known shapes and those novel ones with a shared part space, thus alleviating the

---

[*]Haiyong is the corresponding author.

37th Conference on Neural Information Processing Systems (NeurIPS 2023).

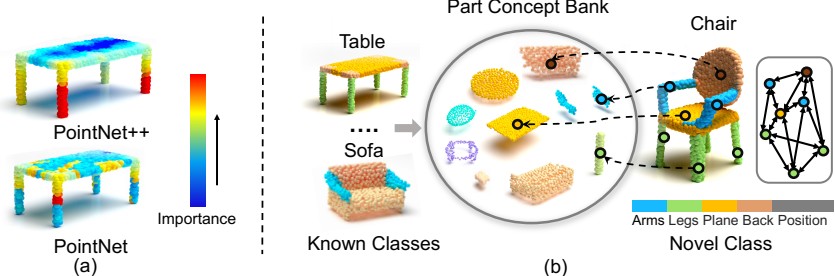

Figure 1: (a) The Grad-CAM [31] of a known shape (table). 3D recognition methods focus on shortcut features representing a special global structure or local region. (b) Novel shape can be represented as a part concept composition along with part-wise spatial relations.

feature bias problem in 2D NCD extensions. On the other hand, the compositions and spatial relations between parts make it easier to recognize novel shapes. For example, as shown in Fig. 1(b), a novel class, e.g., a chair, contains familiar concepts like plane, leg, and backrest observed in known classes (e.g., tables and sofas), therefore we can easily recognize the chair according to its part concepts and part spatial relations.

Along with this idea, we present a novel part concept-based framework coined *DNIK* (**d**ecompose **n**ovel **i**nto **k**nown parts) for 3D NCD as shown in Fig. 2. Specifically, we first construct a learnable part concept bank that encodes rich geometric patterns from known classes. Then *DNIK* can project translation-invariant part-level features of an input shape into the spanning space of part concepts so that shapes from different categories can be represented as part-concept composition features sharing the common embedding space. Therefore learned features by known class recognition can be generalized to novel classes. We further leverage part-wise spatial relations of input shapes with a part relation encoding module (PRE) for discriminative part relation features. These two kinds of features are then combined for 3D shape recognition. To avoid the collapse of the part concept bank and encourage diverse part concepts across classes, three constraints on part concepts are also considered. Moreover, we construct three 3D NCD tasks for the evaluation of different scenarios and also extend our method to the generalized class discovery task (GCD) [35].

Our contributions can be summarized as follows: **(i)** the analysis of biased features and the necessity of part concepts in 3D NCD; **(ii)** a part concept-based framework leveraging both part concept features and part-wise relations for 3D NCD; **(iii)** the design of three necessary constraints on part concepts to facilitate effective part concept learning; **(iv)** various 3D NCD and GCD tasks for the evaluation of different scenarios. Extensive experiments show that our method significantly outperforms baseline methods on all 3D NCD and GCD tasks and can efficiently recognize novel shapes.

## 2 Related Work

**Novel Class Discovery.** As there is no existing work on 3D NCD, we mainly review related works on 2D NCD. Related works can be classified into two categories. The first category of methods [9, 14, 15] usually first train a classification network on the known dataset and then cluster the novel data according to the prediction or output features of the pre-trained network. However, learned features in this way are highly biased toward known classes, leading to performance bottlenecks in novel classes. By contrast, the other line of works [7, 10] explores both known and novel samples for NCD during the clustering phase. Therefore feature representation can be shared between known and novel classes and allow for better generalization to novel classes. AutoNovel [10] utilizes both known and novel samples for self-supervised representation learning and joint known class classification and NCD. Following works further bridge the gap between known and novel classes by mixing up known and novel samples [49], and introducing consistency between similar objects [48] and predictions from two different branches [44]. On the other hand, UNO [7] unifies the learning of both known and novel classes using a single cross-entropy loss based on optimal transports between pseudo-label assignments. Following this paradigm, the similarity relation of samples between known and novel classes is further explored by a spacing loss in the embedding space [16], compositional experts [41], and symmetric KL divergence [19]. Recently NCD has been extended to the more practical general

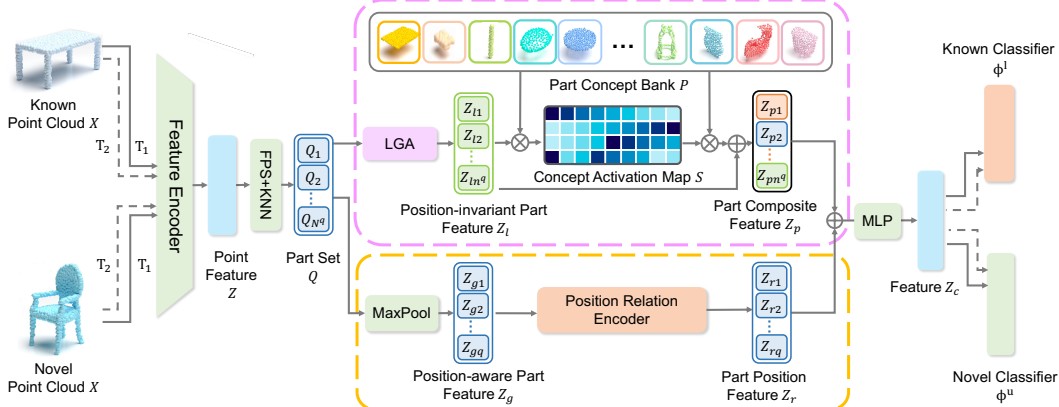

Figure 2: **The overall architecture**. A point cloud and its augmentation are fed into a backbone network to encode point-wise features $Z$ and grouped into part set $Q$ respectively. As illustrated in the pink box, we propose a **local geometric aggregation (LGA)** module to learn position-invariant part feature $Z_l$ and calculate the concept activation map $S$ use the **Poincaré distance** between $Z_l$ and learnable **part concept bank** $\mathcal{P}$. Then we can construct part composite features $Z_p$ according to the Eq. 2. We also propose a **position relation encoder (PRE)** module to extract the part position feature $Z_r$ as illustrated in the orange box. Finally, $Z_p$ and $Z_r$ are concatenated and fed into an MLP for recognition.

category discovery task (GCD) [35], which assumes that unlabeled datasets contain both known and novel classes. [35] and [39] tackle this problem by using semi-supervised K-means and purely parameterized models respectively. In this work, we address 3D NCD and GCD by introducing part concepts and part-wise relations as the bridge between known classes and novel ones.

**Part-based 3D Recognition.** Shape parts are the basic building blocks for 3D shapes [22, 4] and play an important role in both human perception and various 3D recognition tasks. For example, Zhao et al. [46] represent repeatable shape parts as prototypes to improve few-shot 3D object detection. Chowdhury et al. [5] learn part information as a robust prior for incremental learning. Wei et al. [38] align part-based features to reduce the geometry shifts across different domains. Zhao et al. [47] propose a part codebook-based self-attention to leverage the generalized geometric information for better voxel-based feature extractors. Inspired by these works, we learn a part concept bank to map features of known and novel classes into a shared space.

## 3 Method

This work aims at 3D NCD that learns to group 3D shapes in the unlabeled set into several classes by leveraging the knowledge of known classes in a labeled dataset. Basically, 3D NCD assumes the availability of two datasets: an unlabeled dataset $D^u = \{X_i^u\}_{i=1}^{N^u}$ with 3D shapes sampled from novel classes $C^u$ and a labeled dataset $D^l = \{(X_i^l, y_i^l)\}_{i=1}^{N^l}$, where $X_i^l$ denotes a 3D shape and $y_i^l \in C^l$ is the corresponding categories label. The classes in the two datasets are disjoint, therefore we have $C^l \cap C^u = \emptyset$. However, shapes from $D^l$ and $D^u$ usually endow with some similarities. For example, both a chair and a desk have legs and plane surfaces. Following the literature [7, 10, 48], we assume the number of novel classes $C^u$ is known a prior.

### 3.1 Analysis on the Extension of 2D NCD Methods

As there is no existing work on 3D NCD, we instead extend a mainstream 2D NCD method UNO [7] to the 3D domain as the baseline. UNO learns a shared feature extractor $f$ and two linear classifiers $\phi^l$ and $\phi^u$ with $C^l$ and $C^u$ output neurons. The classifiers $\phi^l$ and $\phi^u$ are optimized with a cross-entropy loss using labels of known classes and pseudo labels of novel classes generated by a clustering method (*i.e.*, Sinkhorn-Knopp [6]). However, naive extensions of 2D NCD to the 3D domain work badly for two reasons. First, feature representation is usually dominated by known classes, while oblivious to novel classes. Second, shape-level features extracted from 3D networks confront serious

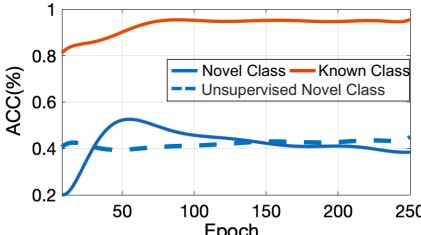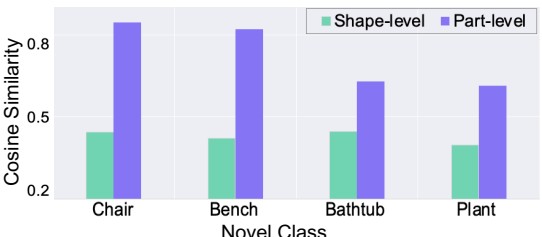

Figure 3: **Left:** The training curves of UNO [7] based methods. **Right:** Cosine similarities between features of novel classes and shape or part-level concepts.

shortcuts as illustrated in Fig. 1. Thus learned features are heavily biased toward the known classes and are hard to serve novel classes.

We conduct a toy experiment to verify our analysis. We chose $table, sofa, stool$ as known classes and $chair, bench, bathtub, plant$ as novel classes from ModelNet [40]. The first two novel classes have similar shape parts with the known classes, e.g., legs, while the last two do not. To reveal the wane and wax between known and novel classes during optimization, we investigated performance curves of UNO [7] in Fig. 3 Left. In the first 60 epochs, training 3D recognition with known class supervision can help novel class discovery. Therefore there exist some sharing feature representations between two different sets of classes and UNO can achieve better accuracy on novel classes than unsupervised clustering (*i.e.*, w/o the cross-entropy loss). However, with the training proceeding, the performance in novel classes starts to decrease and can even be worse than unsupervised clustering, indicating that learned features are gradually biased to known classes and lose generalizability [32].

Fig. 3 Right shows the average cosine similarity between the shape-level or part-level concepts of known classes and the shape or part features of novel classes learned from the trained UNO method [7]. Shape features are direct outputs of the feature extractor while part features denote pooling features of overlapped parts split from shapes. We then define the shape-level concepts by averaging the shape features of each known class and part-level concepts by clustering the part features of known classes using K-means. For shape-level concepts, the cosine similarities of chairs and benches are low and comparable to those of bathtubs and plants, which indicates that shape features cannot be shared between known and novel classes. However, part-level concepts can provide higher cosine similarities and reflect similar properties between known and novel classes better.

## 3.2 Part Concepts for Cross-category Generalization

In this work, we approach 3D NCD with a learnable part concept bank shared across different classes. The overall framework is shown in Fig 2. The basic observation is that geometric patterns and shape parts, *i.e.*, part concepts, are more likely to occur in different categories of shapes as analyzed in Sec. 3.1. We first learn part concepts from shapes of known classes in order to deliver more generalizable feature representation and reduce biases when facing novel classes. Then we leverage the relative positions of different parts to further enhance the recognition of novel classes.

**Learning Part-level Features.** As we intend to learn a set of part concepts representing generalizable shape parts, we first encode part-level features inspired by the paradigm of PointMAE [23]. Specifically, we extract point-wise features $Z \in \mathbb{R}^{L \times D}$ for each point in the input point cloud $X \in \mathbb{R}^{L \times 3}$ with an off-the-shelf 3D backbone, e.g., PointNet [24], where $L$ denotes the number of 3D points and $D$ is the dimension of point cloud features. Then we use farthest point sampling (FPS) to sample $N^q$ points as the centroid seeds and group $K$ neighboring points as overlapped parts $Q = \{Q_i\}_{i=1}^{N^q}$. Part set $Q_i = \{z_{i1}, \cdots, z_{iK}\} \in \mathbb{R}^{K \times D}$ includes features at its centroid $z_{i1}$ followed by point features of its $K-1$ nearest neighbors $\{z_{i2}, \cdots, z_{iK}\}$. Intuitively, we can obtain part-level features by pooling point features within each part $Q_i$, but part features obtained in this way are dependent on the part location and may hinder the learning of shared part concepts at different positions. To cope with this problem, we propose a local geometric aggregation (LGA) module following [42, 26] to decouple the spatial location and aggregate part-level features $Z_l \in \mathbb{R}^{N^q \times D}$ by

$$Z_{l_i} = \psi(\text{maxpool}(\frac{Q_i - z_{i1}}{\sigma})), \forall i \in \{1, \cdots, N^q\}, \tag{1}$$

where $\sigma$ is the scalar standard deviation across channels, and $\psi$ denotes a multi-layer perception (MLP) with channels $(D \times \frac{D}{2} \times D)$ to encode the local geometry features.

**Part Concept learning.** After obtaining part-level features $Z_l$, we use them to learn shared part concepts among categories. First, we construct a learnable part concept bank denoted with $\mathcal{P} = \{P_m\}_{m=1}^M$, where part concept $P_m \in \mathbb{R}^{1 \times D}$ encodes features of different shape parts and can represent *a plane, corners, legs* as illustrated on the top of Fig. 2. In our implementation, part concepts are initialized with a Gaussian distribution $N(\mathbf{0}, \mathbf{I})$. We project part-level features $Z_l$ into the spanning space of part concepts $P$ according to their respective similarities. Then part-level features $Z_l$ are approximated by part composition features $Z_p \in \mathbb{R}^{N_q \times D}$ as follows:

$$Z_{p_i} = \sum_j S_{ij} P_j, \tag{2}$$

where $S \in \mathbb{R}^{N^q \times M}$ measures the similarity between part features $Z_l$ and the part concept bank $P$. The concept activation map $S$ is defined as:

$$S = h(d(Z_l, P^T)), Z_l \leftarrow W_z \cdot Z_l, P \leftarrow W_p \cdot P, \tag{3}$$

where $W_z, W_p \in \mathbb{R}^{D \times D_p}$ are learnable weight matrixes for feature dimension reduction, $h(\cdot)$ is a softmax function to produce normalized probabilities along each row, and $d(\cdot, \cdot)$ is a distance function that compares part features and part concepts. To ensure that the geometric knowledge of each part feature can be learned by a particular part concept, distance $d(\cdot, \cdot)$ should produce significant similarity differences between part features and different part concepts. A common choice of $d(\cdot, \cdot)$ is to use dot product similarity [38] or cos similarity [11] defined in the Euclidean space. However, as shown in Fig. 4(a), we found that the Euclidean distance functions lead to uniform activations of part features on different part concepts. Different from the Euclidean space where the distance grows linearly, the distance between part features and concepts grows exponentially in the hyperbolic space [17, 30, 28], which can produce sharp concept activation maps (see Fig. 4(a)). Therefore, we project features into the hyperbolic space using Poincaré sphere model following [17] and the distance function between $Z_l$ and $P$ in Poincaré manifolds is given by:

$$d(Z_l, P) = \text{arcosh}(\frac{2\|Z_l - P\|_2}{(1 - \|Z_l\|_2)(1 - \|P\|_2)} + 1). \tag{4}$$

Though learned part concepts can cover most shape parts in novel classes, there still exist some novel part concepts that are not contained in known classes. In this case, the concept activation map between a part and part concepts $P$ is likely to be uniform and results in ambiguous composite features. In our method, we further concatenate the local part features $Z_l$ with the part composite features $Z_p$ to confront this problem.

**Part Relation Encoder.** As part composite features $Z_p$ only consider position-independent compositions of part concepts and neglect mutual position relations between different parts that usually deliver shape structures as shown in Fig. 1(b). To this end, we propose a novel Position Relation Module that adopts a shallow Transformer Encoder [45] with two self-attention layers, denoted as TransEnc, to learn mutual relations between part concepts as relation feature $Z_r$. Since the position information has been decoupled from $Z_l$, we regain position-aware part features $Z_g = \{Z_{g_i}, ..., Z_{g_{N_q}}\} \in \mathbb{R}^{N_q \times D}$ from $Q$ using max-pooling and encode $Z_r \in \mathbb{R}^{N_q \times D}$ as follows:

$$\begin{aligned} Z_{g_i} &= \text{maxpool}(Q_i), \\ Z_r &= \text{TransEnc}(Z_g). \end{aligned} \tag{5}$$

Afterward, we concatenate the position relation feature $Z_r$ with the point composite features $Z_p$ and feed the result into an MLP with $2D \times D$ and a max-pooling layer to produce category-aware features $Z_c$. We can classify the input shape $X$ by feeding $Z_c$ to a linear layer predicting the class-wise probabilities.

### 3.3 Constraining Part Concepts

Though part concepts are proposed to represent different parts, they can easily collapse to the same values. The main reason is that the similarity between randomly initialized part concepts and part

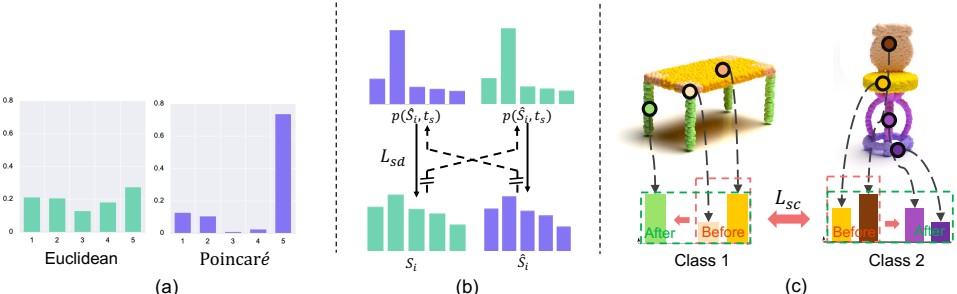

Figure 4: (a) The part concept activation is likely to be uniform in the Euclidean space but sharp in the Poincaré space. (b) Self-distillation loss $L_{sd}$ encourages part activations to align with the sharper distributions of their corresponding data augmentation. (c) Supervised contrastive loss $L_{sc}$ forces the part concept activations of different known classes to be separated from each other so that more part concepts can be learned.

features is similar during the initial training stage. Consequently, part composite features based on the concept activation map would gradually be identical and part concepts updated with similarly back-propagated gradients from them converge to similar features. To ensure the representation ability and generalization of the part concept bank, we propose three constraints to train part concepts.

**Self-distillation Loss.** To guarantee that each part's geometric knowledge can be learned by specific part concepts, the concept activation distribution of each part should be sharp. To achieve this, we propose a self-distillation loss inspired by [3]. We enforce each part concept activation $S$ of the shape parts $Q$ to be consistent with a sharper one of the corresponding part concept activation $\hat{S}$ from the augmented shape parts $\hat{Q}$ (see Fig 4 (b)) as follows:

$$L_{sd} = -\frac{1}{2} \Big( \text{stop}(p(S_i, \tau_s)) \log \hat{S}_i + \text{stop}(p(\hat{S}_i, \tau_s)) \log S_i \Big), \tag{6}$$

where $p(S_i, \tau_s) = \frac{\exp(s_i/\tau_s)}{\sum_{j=1}^{N^q} \exp(s_i^j/\tau_s)}$ is a sharp concept activation distribution of $Q_i$ controlled by temperature parameter $\tau_s$ (set 0.1 empirically) and stop$(\cdot)$ means the gradients is not back-propagated. Loss $L_{sd}$ enhances the sharpness and consistency of the concept activation under different data augmentations for both the labeled and unlabeled datasets.

**Supervised Contrastive Loss.** During training, we found that even a few part concepts can support the recognition of known classes as shown in the red boxes in Fig. 4(c). In this case, the part concept bank lacks the incentive to learn more part concepts from known shapes, which reduces the information it can share with novel classes. Thus, for the labeled dataset $D^l$, we employ a supervised contrastive loss $L_{sc}$ to increase the inter-class discrepancy of their concept activation maps. As shown in the green boxes in Fig. 4(c), through $L_{sc}$, the part concept bank can learn additional parts (*i.e.*, leg and pedestal) from known shapes to increase their inter-class discrepancy. Given a mini-batch of $B$ shapes $\{X_b\}_{b=1}^B$ and their augmentation $\{\hat{X}_b\}_{b=1}^B$, we first sum their activation maps $S_B \in \mathbb{R}^{2B \times M \times N^q}$ into shape concept activation $T_B \in \mathbb{R}^{2B \times M}$ along the part dimension, where $T_i$ summarizes all part concepts in a shape. Then the supervised contrastive loss is defined on $2B$ shapes as follows:

$$L_{sc} = -\frac{1}{|\rho|} \sum_{T_j^+ \in \rho} \log \frac{\exp(-\text{sim}(T_i, T_j^+)/\tau_c)}{\sum_{b=1}^{|B|} \exp(-\text{sim}(T_i, T_b)/\tau_c)}, \tag{7}$$

where $\rho = \{\{T_j^+ \in T_B : y_j = y_i\} \cup \hat{T}_i\}$ is the set that contains the shape concept activation $\hat{T}_i$ of the augmented sample and other samples belonging to the same class in the mini-batch, sim$(.,.)$ calculates the cosine similarity, and temperature $\tau_c$ is set 0.1 empirically.

**Diversity Loss.** To capture different patterns in 3D shapes, part concepts should be diverse. We can encourage this by minimizing the similarity between different part concepts with the following loss:

$$L_{cd} = \sum_{m=1}^M \max_{P^- \in \{P_m\}_{m=1}^M/P_m} (0, \text{sim}(P_m, P^-) - \delta), \tag{8}$$

where $\delta$ is the cosine similarity threshold between a code and the other ones. In our experiments, we set $\delta = 0.1$.

## 3.4 The Training Losses

Following the literature [48, 10, 7], we construct two linear classifiers $\phi^l$ and $\phi^u$. We employ a standard cross-entropy loss $L_{ce}$ to optimize the known class classifier $\phi^l$ and a self-classification loss $L_{self}$ for training novel class classifier $\phi^u$. Loss $L_{self}$ minimizes the cross entropy between novel shapes and their augmentation with the assumption that the distribution of each novel class is uniform and is formulated following [2]. In summary, the overall training objective loss function in our model is:

$$L = L_{ce} + L_{self} + \lambda_1 \cdot L_{sd} + \lambda_2 \cdot L_{sc} + \lambda_3 \cdot L_{cd}, \tag{9}$$

where loss $L_{sd}$ in Eq.(6), $L_{sc}$ in Eq.(7) and $L_{cd}$ in Eq.(8) enforce basic assumptions of part concepts. Weight term $\lambda_*$ balances different losses and is set to $(0.1, 0.1, 0.1)$. During inference, part composite features of a test sample $Z_c$ are passed through both classifiers to obtain the corresponding logits, then the two logits are concatenated and fed into a softmax layer to obtain the class-wise probability.

## 4 Experiments

### 4.1 Evaluation Datasets for 3D NCD

We construct three NCD tasks to evaluate the performance of 3D NCD based on four different kinds of 3D datasets covering CAD shapes (*i.e.*, ModelNet40 [40] and ShapeNetCore [4]), and indoor objects constructed using 3D scans and multi-view images (*i.e.*, ScanObjectNN [33] and Co3D [27]).

**Random Split 3D NCD** focuses on evaluating the impact of a different number of known and novel classes on the 3D NCD following [7]. The task is denoted with *DATASET-K-U*, where $K$ is the number of known classes and the other $U$ classes as novel ones.

**Similarity Split 3D NCD.** The semantic similarity between known and novel classes can greatly affect the performance of NCD. A highly semantically similar label set can provide more useful information for discovering novel classes. Based on the transfer leakage metric in [20], we split novel classes as three sets (i.e., *High, Medium, and Low*) according to the descending similarity w.r.t known classes.

**Cross Domain 3D NCD** assesses the performance of 3D NCD on cross-domain shape recognition and reflects the real-world scenarios where novel shapes may come from different domains. Therefore, we select known and novel classes with semantic similarity from different datasets respectively. The corresponding tasks are denoted with (known class dataset $\rightarrow$ novel class dataset).

### 4.2 Comparison with SOTA methods

To our best knowledge, there is no previous attempt at the 3D NCD problem. So we extend different kinds of SOTA 2D NCD methods to 3D NCD, including *DTC* [9], *AutoNovel* [10], *NCL* [48], *UNO* [7], and *IIC* [19]. In addition, training a novel class classifier with pseudo labels generated by K-means can achieve good results and we also compare with it, denoting as *Kmeans+*. We implement the above baselines by substituting the image-based backbone with a 3D one (*i.e.*, PointNet) for 3D NCD.

**Results on the Random Split 3D NCD Task** are shown in Tab. 1. For the known class recognition, all compared methods perform well, but the performance on the novel classes diverges a lot. When the number of known classes is large, the network can inherit more knowledge for the novel class recognition. The NCL and IIC can achieve better results on most datasets since they force the separation between known and novel classes while learning similarity relationships within the novel classes, so the learned features are more discriminative for the novel classes. Our method can gain consistent improvements on novel classes of all tasks ($+11.7\%$ on average) while attaining the best know-class accuracy for most tasks ($5/6$). This demonstrates that part concepts can effectively bridge the gaps between known and novel shapes thus mitigating the feature bias. When the number of known classes is small, the performance of compared methods drops significantly. By contrast, our method decreases much less and still achieves the best results ($+12.5\%$ increases on average). We

Table 1: The quantitative results of the random split 3D NCD task. The results are averaged among five random trials. The best results are in **bold**, and the second-best ones are underlined.

| | ModelNet 30-10 | | ModelNet 10-30 | | Co3D 30-20 | | Co3D 20-30 | | ShapeNet 40-15 | | ShapeNet 20-35 | |
| --- | --- | --- | --- | --- | --- | --- | --- | --- | --- | --- | --- | --- |
| | Novel | Known | Novel | Known | Novel | Known | Novel | Known | Novel | Known | Novel | Known |
| DTC [9] | 49.7 | 79.6 | 31.1 | 92.5 | 31.1 | 77.0 | 21.9 | 84.6 | 40.8 | 80.8 | 31.0 | 89.9 |
| AutoNovel [10] | 46.6 | 88.1 | 46.7 | 93.8 | 52.2 | 80.6 | 42.0 | 86.1 | 36.1 | 82.2 | 35.2 | 91.4 |
| NCL [48] | 61.3 | **89.1** | 51.2 | 93.7 | 52.4 | 80.2 | 39.5 | 89.6 | 44.4 | 83.4 | 42.2 | 87.7 |
| UNO [7] | 59.3 | 86.7 | 45.4 | 93.2 | 48.6 | 80.1 | 41.6 | 87.3 | 45.5 | 82.2 | 28.6 | 90.7 |
| IIC [19] | 62.7 | 87.6 | 49.7 | 94.2 | 45.9 | 80.0 | 37.4 | 86.7 | 44.3 | 81.5 | 29.1 | 90.6 |
| Kmeans+ | 59.9 | 83.0 | 41.6 | 92.2 | 52.1 | 76.6 | 43.8 | 84.8 | 48.1 | 80.9 | 36.6 | 90.9 |
| Ours | **72.0** | 88.9 | **63.0** | **96.0** | **61.0** | **85.0** | **58.5** | **90.1** | **62.6** | **84.7** | **53.2** | **92.4** |
| Improvement | +9.3 | -0.2 | +11.8 | +1.3 | +8.6 | +4.4 | +14.7 | +0.5 | +14.5 | +1.3 | +11.0 | +1.0 |

Table 2: The accuracy results on the similarity split 3D NCD task. *High, Medium, Low* denote the novel classes have *High, Medium, Low* semantic similarity with known classes.

| | ModelNet | | | | Co3D | | | | ShapeNet | | | |
| --- | --- | --- | --- | --- | --- | --- | --- | --- | --- | --- | --- | --- |
| | High | Medium | Low | Known | High | Medium | Low | Known | High | Medium | Low | Known |
| DTC [9] | 28.3 | 39.6 | 32.5 | **95.7** | 41.3 | 35.2 | 26.7 | 89.2 | 37.6 | 36.6 | 36.6 | 87.5 |
| AutoNovel [10] | 39.2 | 42.2 | 44.6 | 94.8 | 57.1 | 54.7 | 34.7 | 87.5 | 39.2 | 38.3 | 39.2 | 87.1 |
| NCL [48] | 56.6 | 56.5 | 45.1 | 95.2 | 57.4 | 53.1 | 39.7 | 89.8 | 45.3 | 36.9 | 38.7 | 87.1 |
| UNO [7] | 49.8 | 47.4 | 45.5 | 95.1 | 49.4 | 48.1 | 31.3 | 87.9 | 35.4 | 43.9 | 31.2 | 85.4 |
| IIC [19] | 59.6 | 56.0 | 38.9 | 93.9 | 55.8 | 48.8 | 43.1 | 84.5 | 47.1 | 38.4 | 28.7 | 85.6 |
| Kmeans+ | 58.5 | 54.2 | 39.6 | 95.0 | 54.3 | 49.5 | 37.3 | 84.6 | 44.9 | 45.9 | 33.8 | 86.1 |
| Ours | **73.2** | **68.3** | **66.4** | 95.2 | **69.3** | **65.2** | **56.9** | **93.3** | **64.7** | **54.6** | **57.1** | **89.2** |
| Improvement | +13.6 | +11.8 | +20.9 | -0.5 | +11.9 | +10.5 | +13.8 | +3.5 | +17.6 | +8.7 | +17.9 | +1.7 |

conjecture that the part concept bank can learn more local features for sharing than the other methods using global features.

**Results on the Similarity Split 3D NCD Task** are presented in Tab. 2. *NCL, UNO, IIC, and Kmeans+* can achieve better results on novel classes with high similarities. However, the performance of these methods drops significantly ($-13.5\%$ on average) with the decrease in semantic similarity. The method that achieves the best results on *High*-similarity novel classes cannot obtain the best on *Medium* and *Low* ones simultaneously. This indicates that the learned knowledge of these methods is biased towards known classes with highly similar geometry while lacking generalizability to other novel ones. By contrast, our method achieves large improvements for all similarities ($+14.3\%$, $+11.8\%$, $+17.5\%$ on average) while reducing the gap between different similarity tasks. This is because part concepts can show up in different shapes, e.g., the back of a sofa and a chair in Fig. 1 and align all shapes into the part space, allowing for generalization even for low-similarity novel classes.

**Results on the Cross-Domain 3D NCD Task** are reported in Tab. 3. We can observe that compared methods lack robustness on cross-domain shapes and perform poorly. Our method performs well among different datasets and outperforms existing methods by a large margin ($+16.3\%$ on average). We believe the good performance is owing to the fact that part concepts are robust to data domains and can be generalized to different kinds of data.

### 4.3 Ablation Study

In this part, we conduct all ablations on the ModelNet similarity split 3D NCD. Readers can find more experiments in the supplementary.

**Model components.** In Tab. 4-②, adding the part concept bank without the LGA only achieves a very small improvement compared to the baseline, which indicates part concepts without positional invariance are hard to fit novel classes. In ③ and ④, LGA can ensure better generalization over novel classes, and the part concept bank can align the part features from known and novel classes into the same space. Therefore they can collaborate to boost the performance for a large margin ($5.8\%$ and $3.9\%$). In ⑤, using the Poincaré metric can yield much sharper concept activations and make different part concepts more discriminative, therefore raising the performance ($+3.7\%$ and $+8.5\%$). In ⑥, self-distillation loss $L_{sd}$ encourages each part concept to learn specific patterns to avoid collapses and contributes to an average $6.25\%$ gain in accuracy. In ⑦ and ⑧, loss $L_{sc}$ and $L_{cd}$ force the network to

Table 3: The results on cross-domain task. M. and Scan. are ModelNet and ScanObjectNN.

| | M.→Co3D | M.→Scan. | Co3D→Scan. |
|---|---|---|---|
| | Novel | Novel | Novel |
| DTC [9] | 26.6 | 29.6 | 25.2 |
| AutoNovel [10] | 26.8 | 36.7 | 41.6 |
| NCL [48] | 35.2 | 39.0 | 37.4 |
| UNO [7] | 27.7 | 43.9 | 35.0 |
| IIC [19] | 30.8 | 39.8 | 35.4 |
| Kmeans+ | 29.0 | 41.8 | 26.0 |
| Ours | **64.0 +28.8** | **53.4 +9.5** | **52.5 +10.5** |

Table 4: Ablation experiments. PCB means part concept bank. LGA, Poincaré, and PRE are in Fig. 2.

| | LGA | PCB | Poincaré | $L_{sd}$ | $L_{sc}$ | $L_{cd}$ | PRE | High | Low |
|---|---|---|---|---|---|---|---|---|---|
| ① | | | | | | | | 48.8 | 37.9 |
| ② | | ✓ | | | | | | 49.9 | 40.3 |
| ③ | ✓ | | | | | | | 52.6 | 44.8 |
| ④ | ✓ | ✓ | | | | | | 58.4 | 48.7 |
| ⑤ | ✓ | ✓ | ✓ | | | | | 62.1 | 57.2 |
| ⑥ | ✓ | ✓ | ✓ | ✓ | | | | 67.9 | 63.9 |
| ⑦ | ✓ | ✓ | ✓ | ✓ | ✓ | | | 70.1 | 64.8 |
| ⑧ | ✓ | ✓ | ✓ | ✓ | ✓ | ✓ | | 71.3 | 65.1 |
| ⑨ | ✓ | ✓ | ✓ | ✓ | ✓ | ✓ | ✓ | **73.2** | **66.4** |

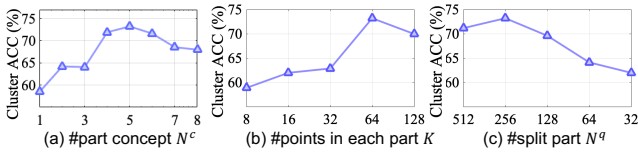

Figure 5: Hyperparameter evaluations on (a) the number of part concepts, (b) the number of points $K$ contained in each part, and (c) the number of split parts $N^q$ of each shape.

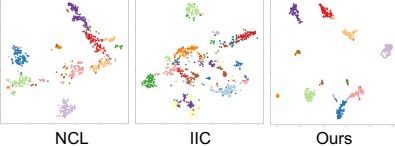

Figure 6: Visualization of the embedding spaces of novel classes with t-SNE for different methods.

learn more diverse part concepts, which is more helpful for high-similarity novel classes ($3.4\%$ vs $1.2\%$), since their concept activation maps are more similar (*e.g.*, chair and stool). By comparing ⑤ and ⑧, we can see the three proposed constraints ($L_{sd}$, $L_{sc}$, and $L_{cd}$) are essential to the final good performance and strengthen the results by $+8.8\%$ and $+7.9\%$. In ⑨, the PRE can further incorporate discriminative position relation features to enhance the performance ($+1.6\%$ on average).

**Hyperparameters.** In Fig. 5(a), we conduct experiments on different numbers of part concepts, *i.e.*, $M$. Considering part concepts should increase with known classes, we set $M = N^c \times C^l$, When $N^c \leq 3$, the results are poor, which indicates that few part concepts cannot adequately encode all part patterns in known classes. However, the performance also drops when $N^c > 6$. This may be because too many concepts are harder to be optimized.

The number of points contained in each overlapped part determines the part size, therefore affecting the generalization and discrimination abilities. The discrimination ability of parts decreases when the size is reduced and the generalization ability improves. Specifically, small parts are hard to contain meaningful local geometric patterns, while large parts may contain special structures that are biased towards known classes as shown in Fig. 5(b). Our approach achieves the best results when $K = 64$.

Since the part set of a shape is split by FPS, few parts lead to large variations in the part sets of similar shapes, resulting in inconsistent part composite features. In Fig. 5(c), the result will decrease continuously when $N^q < 128$. Our method achieves the best performance when $N^q = 256$.

**Visualizations of the Feature Embedding** with t-SNE [34] are shown in Fig. 6. For the NCL [48] and IIC [19], the features of different novel classes are mixed up in the embedding space, while ours can push away embedding regions of different novel classes to a greater extent.

**Part Concept Visualization.** In order to reveal what part concepts can learn, we show concept activation maps between some part concepts and part features in Fig. 7. We can see that part concepts can activate similar semantic parts of both known and novel shapes, such as legs, arms, bottles, backrests, planes, and screens. This confirms that part concepts can effectively generalize knowledge learned from known classes to novel classes and represent novel classes as the composition of part concepts. For example, a chair can be represented as {legs, arms, a backrest, and a plane}. We notice that part concepts can be robustly discovered and activated even if geometric structures are slightly different, e.g., arms of chairs, sofas, and benches.

## 4.4 Extension for Generalized Class Discovery

*DNIK* can also be extended for generalized class discovery (GCD) [35], for which the unlabeled dataset $D^u$ contains 3D shapes from both known and novel classes. In order to adapt *DNIK* for this task, we concatenate the known class classifier $\phi^l$ and the novel class classifier $\phi^u$ into a unified

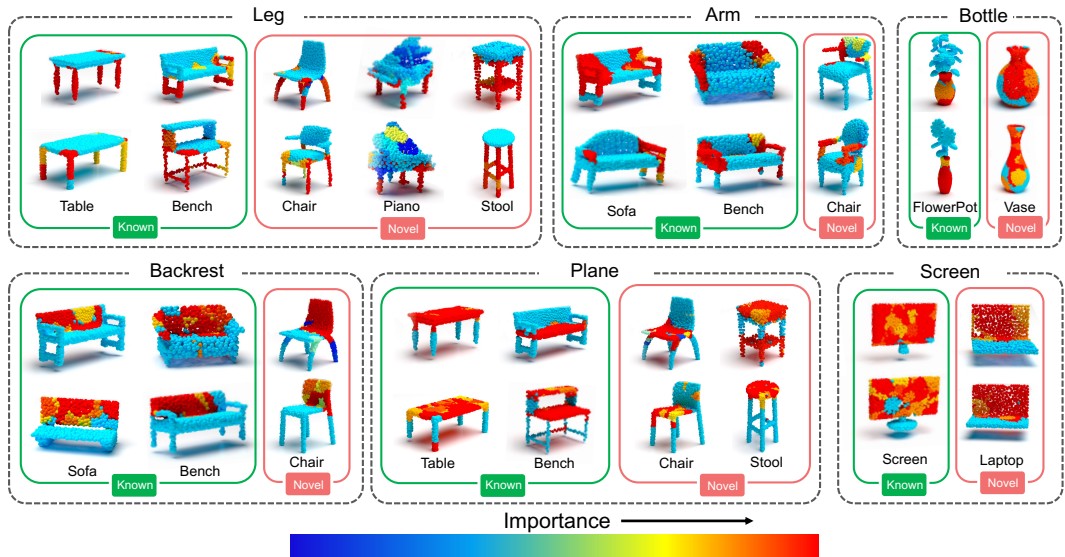

Figure 7: Visualization of different part concepts and their corresponding activations on different shapes of known and novel classes. Red indicates higher activations and similarities.

Table 5: The quantitative results of the random split 3D GCD task.

| | ModelNet 30-10 | | ModelNet 10-30 | | Co3D 30-20 | | Co3D 20-30 | |
|---|---|---|---|---|---|---|---|---|
| | Novel | Known | Novel | Known | Novel | Known | Novel | Known |
| AutoNovel+ [10] | 38.6 | 84.7 | 34.9 | 76.5 | 33.1 | 78.9 | 34.2 | 84.4 |
| UNO+ [7] | 42.1 | 82.1 | 39.3 | 87.2 | 45.2 | 73.8 | 43.3 | 85.9 |
| GCD [19] | 50.6 | 89.2 | 49.2 | 89.5 | 46.4 | 77.3 | 45.8 | 87.3 |
| SimGCD [39] | 44.8 | 89.0 | 41.9 | 87.8 | 39.7 | 78.8 | 33.3 | 85.0 |
| Ours | 69.1 | 88.9 | 68.4 | 92.1 | 56.7 | 83.2 | 54.1 | 87.0 |
| Improvement | +18.5 | -0.1 | +19.2 | +2.6 | +10.3 | +4.3 | +8.3 | -0.3 |

Table 6: The accuracy results on the similarity split 3D GCD task.

| | ModelNet | | | Co3D | | |
|---|---|---|---|---|---|---|
| | High | Low | Known | High | Low | Known |
| AutoNovel+ [10] | 32.6 | 33.6 | 94.3 | 50.8 | 32.1 | 89.6 |
| UNO+ [7] | 41.7 | 36.5 | 88.5 | 52.1 | 41.9 | 88.8 |
| GCD [19] | 58.3 | 54.7 | 95.7 | 56.2 | 46.0 | 89.2 |
| SimGCD | 47.2 | 43.5 | 94.0 | 54.3 | 42.1 | 82.4 |
| Ours | 70.6 | 68.2 | 95.9 | 62.5 | 53.9 | 88.0 |
| Improvement | +12.3 | +13.5 | + 0.2 | +6.3 | +7.9 | -1.6 |

classifier. All samples from both the labeled dataset and unlabeled dataset go through the unified classifier to obtain the class-wise probability. Both $L_{ce}$ and $L_{self}$ are now used to optimize the unified classifier rather than the novel class classifier of the NCD setting.

We extend two 2D GCD methods to 3D GCD, including *GCD* [35], *SimGCD* [39], and also extend AutoNovel [10] and UNO [7] for GCD tasks (denoted as AutoNovel+ and UNO+) by concatenating logits of known classes and novel classes following SimGCD [39]. Results are shown in Tab. 5 and Tab. 6. It can be seen that all compared methods show decreased results compared to the NCD tasks in Tab. 1 and Tab. 2. This indicates that samples of known classes in the unlabeled dataset $D^u$ introduce noise into the training and confuse the classifier. GCD [35] can achieve better novel class accuracy (+5.6%) than other compared methods because it uses semi-supervised Kmeans to cluster overall samples, which can produce more reliable pseudo labels. Our method can still achieve superior performance (+12.1%) on novel classes of all tasks while attaining comparable know-class accuracy for most tasks. This indicates that the idea of leveraging part compositions for novel class discovery can perform well in both the NCD setting and the GCD setting.

## 5 Conclusion

In this work, we present a novel part concept-based framework *DNIK* for 3D NCD. DNIK leverages part concepts and part-wise relations that are widely shared by both known classes and novel ones. Therefore part concepts and part-wise relations learned from known classes can reinforce the recognition of novel shapes. Comprehensive experiments demonstrate the method overwhelms all baselines consistently and significantly on all metrics. We hope our work can inspire more research on find-grained concept learning and 3D NCD.

**Limitations.** Part concepts in the work are not capable of learning multi-scale geometric parts, thus restricting the representation ability and resulting in sensitive scale parameters in Fig. 5 (b).

## Acknowledgement

This work is supported by the National Natural Science Foundation of China (62271467, U2003109, U21A20515, 62102393, 62206263), China Postdoctoral Science Foundation (2022T150639, 2021M703162), the State Key Laboratory of Robotics and Systems (HIT) (SKLRS-2022-KF-11), and the Fundamental Research Funds for the Central Universities.

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

# A    In-depth Analysis of the Method

**Feature Visualization of Novel Classes.** We show the feature distribution of novel classes learned by all compared methods on the high similarity task of ModelNet in Fig. 8. DTC [9] mingles features of different novel classes together and impedes novel class recognition. The feature distributions of AutoNovel [10], NCL [48], UNO [7] and IIC [19] are less confusing but there is no clear category boundary for different novel classes. Kmeans+ can produce better intra-class consistency and can separate most novel classes. A possible reason is that pseudo labels by K-means clustering on the unlabeled training set can reflect the overall distribution to some extent. However, some shapes are incorrectly grouped into the wrong class due to the poor discrimination of learned features. By exploring the part concept compositions of different novel classes, our method can push away the embedding regions of novel classes to a greater extent compared to other methods and achieve much tighter intra-class representations.

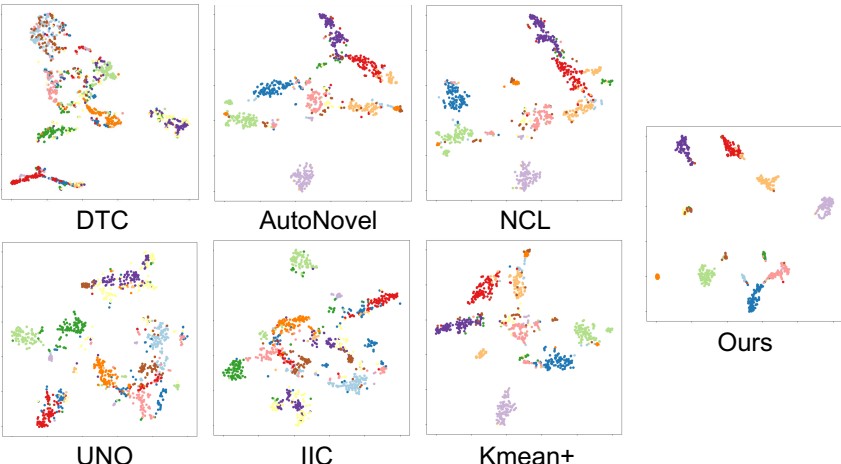

Figure 8: Learned embeddings visualized by 2D t-SNE. We use the ModelNet high-similarity task for the experiment. Each color denotes a different novel class.

**Train Curves.** As described in Sec. 3.1, the learned features of compared methods are gradually biased to known classes during the training process and lose generalizability to novel classes. To further confirm our analysis, we demonstrate the converging curves of known class accuracy and novel class cluster accuracy during the training of seven methods on the high similarity task of ModelNet in Fig 9. As shown in Fig. 9a, the known class accuracy of all methods increases quickly in the first 30 epochs and grows slowly after 100 epochs. However, in Fig. 9b, with the increase in the known class accuracy after 50 epochs, the clustering accuracies for novel classes of UNO, IIC, and Kmeans+ start to decrease. This indicates the known class accuracy competes with the novel class clustering accuracy, and features beneficial to known class recognition instead turn out to harm the generalization to novel classes. The clustering accuracy of our method stays stable after 60 epochs. Meanwhile, the trend of the cluster accuracy is consistent with that of the known class accuracy suggesting that learned features on known classes encourage the recognition of novel ones. This good generalization is attributed to the fact that part concepts construct a shared embedding space between known and novel classes.

**Impact of 3D pre-trained models.** 3D data makes things more challenging as the 3D pretrained model does not generalize well. For example, SimGCD [39] uses the powerful DINO [3] model pre-trained on millions of images as the encoder and only fine-tunes a very small part of the network during training. This training strategy makes the extracted image features more generalizable, thus producing better results. In fact, previous work [29] suggests that DINO is very important to the generalization to unseen concepts. In contrast, the 3D field lacks both large-scale 3D shapes as well as high-quality pre-trained models. In order to evaluate the impact of 3D pre-trained models, we present results using two different self-supervised pre-trained backbones (OcCo [36] and CrossPoint [1]) under the GCD setting in the Tab. 7. Following SimGCD [39], we only fine-tune the last three layers of the PointNet++ backbone and the classification heads. The results demonstrate that utilizing more generalized features from pre-training improves the performance of the baseline methods. However,

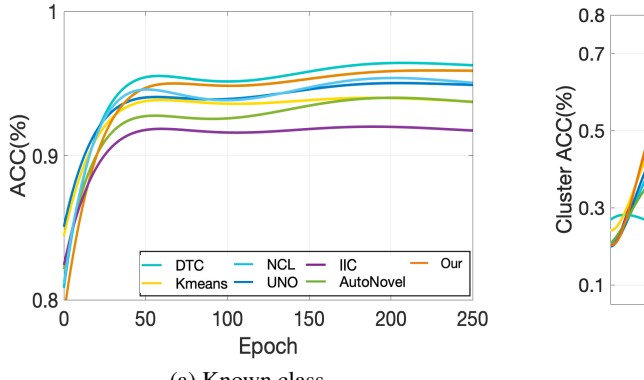 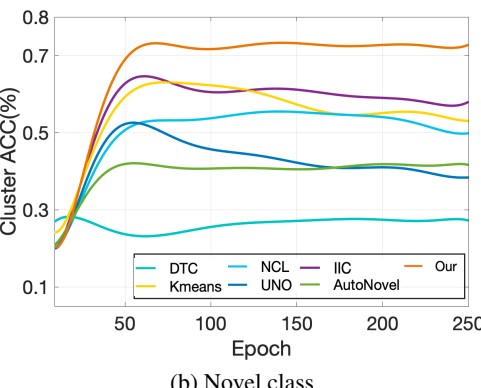

(a) Known class        (b) Novel class

Figure 9: (a) The accuracy curves of known classes during training. (b) The clustering accuracy curves of novel classes during training.

Table 7: Result of using 3D pre-trained models.

|  | Scratch | OcCo [36] | CrossPoint [1] |
|---|---|---|---|
| UNO+ | 41.7 | 42.7 | 45.3 |
| SimGCD [39] | 47.2 | 48.9 | 51.2 |
| Our | 70.6 | - | - |

there remains a substantial gap compared to our method. This indicates that existing 3D pre-trained models have not yet learned representations as powerful as those captured by DINO [3].

## B   Ablation Study

To further investigate our proposed model components, we conduct additional ablation experiments in Tab. 8. In ②, ③ and ④, both Poincareé distance and $L_{sd}$ are proposed to encourage sharper part concept activation. Each of these two modules can be used individually to obtain performance gains (+3.7% vs +8.5%, +5.1% vs +7.4%), and they can be used together to achieve an even bigger improvement (+9.5% vs +15.2%). Without Poincareé distance and $L_{sd}$, using $L_{sc}$ and $L_{cd}$ alone can only achieve minor improvements as shown in ⑤ and ⑥. This is because that part concepts are less diverse and tend to encode the global feature of known classes that are hard to be generalized to novel classes. Since novel shapes may contain new part concepts that are not present in known classes, we concatenate the local part features $Z_l$ with the part composite features $Z_p$. As shown in ⑧, this design can obtain better results on low-similarity novel classes (+0.7% vs + 1.8%). We also evaluate the performance of only using the PRE module as shown in the ⑩. The results show that using PRE alone can already achieve considerable performance gains (4.75% on average).

**Ablation on different known classes number.** Training on more known categories can certainly benefit novel class discovery. As shown in Tab.9, increasing the number of known classes leads to improved performance. Because with multiple categories we can learn more diverse part concepts that further will help us recognize novel objects better. In the table below, we evaluated the performance by training the method with an increasing number of known categories and the fixed number of novel categories. The results show that training the method with more categories does enhance the recognition of novel shapes.

## C   Different Backbones

We inspect how different network backbones impact the performance of 3D NCD. The comparisons are conducted on three popular 3D backbones including PointNet++ [25], DGCNN [37], and Point-NeXt [26]. Among them, the PointNeXt achieves state-of-the-art performance for 3D classification and segmentation. We replace the backbone of compared methods with the above network architec-

Table 8: Ablation experiments. PCB means part concept bank. LGA, Poincaré, Concat and PRE are local geometric aggregation, Poincaré distance, concatenate the local part features $Z_l$ and position relation encoder as shown in Fig. 2.

| | LGA | PCB | Poincaré | $L_{sd}$ | $L_{sc}$ | $L_{cd}$ | Concat | PRE | High | Low |
|---|---|---|---|---|---|---|---|---|---|---|
| ① | ✓ | ✓ | | | | | ✓ | | 58.4 | 48.7 |
| ② | ✓ | ✓ | ✓ | | | | ✓ | | 62.1 | 57.2 |
| ③ | ✓ | ✓ | | ✓ | | | ✓ | | 63.5 | 56.1 |
| ④ | ✓ | ✓ | ✓ | ✓ | | | ✓ | | 67.9 | 63.9 |
| ⑤ | ✓ | ✓ | | | ✓ | | ✓ | | 60.1 | 51.5 |
| ⑥ | ✓ | ✓ | | | | ✓ | ✓ | | 59.8 | 49.3 |
| ⑦ | ✓ | ✓ | ✓ | ✓ | ✓ | ✓ | | | 70.6 | 63.3 |
| ⑧ | ✓ | ✓ | ✓ | ✓ | ✓ | ✓ | ✓ | | 71.3 | 65.1 |
| ⑨ | | | | | | | | | 48.8 | 37.8 |
| ⑩ | | | | | | | | ✓ | 52.4 | 43.7 |
| ⑩ | ✓ | ✓ | ✓ | ✓ | ✓ | ✓ | ✓ | ✓ | **73.2** | **66.4** |

Table 9: The results of different known class number.

| ModelNet | 5-10 | 10-10 | 20-10 | 30-10 |
|---|---|---|---|---|
| Novel Acc | 57.9 | 61.3 | 64.0 | 66.2 |

tures and evaluate their performance on the ModelNet high similarity task. The results are shown in Tab. 10. the known class accuracy of the same 3D NCD method on different backbones does not vary too much (within $1\%$). On the other hand, the backbone has a significant influence on the novel-class performance for all the compared methods. For example, the weaker backbone DGCNN can achieve better novel class accuracy than the other two backbones. This is because DGCNN uses graphs to represent the local features of the point cloud, thus increasing the feature generalization on novel classes. Our method can surpass all the baselines by a very large margin (at least $+8.9\%$) on the novel-class accuracy even using a weak backbone (PointNet), while attaining comparable good performance on the known-class recognition. This superiority is primarily attributed to the part concept-based features that are more generalizable to novel classes than other backbones.

Table 10: The accuracy results of different backbones on the similarity split 3D NCD task. *High, Low* denote the novel classes have *High, Low* semantic similarity with known classes.

| | PointNeXt | | | PointNet++ | | | DGCNN | | |
|---|---|---|---|---|---|---|---|---|---|
| | High | Low | Known | High | Low | Known | High | Low | Known |
| DTC [9] | 43.8 | 32.6 | 93.9 | 50.9 | 44.4 | 94.3 | 51.3 | 38.0 | 94.6 |
| AutoNovel [10] | 51.6 | 46.5 | 96.0 | 59.0 | 43.4 | 96.7 | 49.0 | 40.9 | 96.7 |
| NCL [48] | 59.9 | 46.5 | 95.9 | 51.3 | 37.6 | **97.0** | 64.3 | 44.9 | 95.1 |
| UNO [7] | 61.6 | 32.9 | 95.8 | 53.0 | 43.2 | 93.0 | 62.9 | 45.6 | 96.7 |
| IIC [19] | 62.3 | 49.7 | 95.3 | 57.6 | 46.0 | 94.6 | 64.0 | 51.8 | **96.8** |
| Kmeans+ | 61.2 | 50.9 | **96.4** | 55.2 | 40.7 | 96.2 | 64.0 | 45.5 | 92.3 |
| Ours (PointNet) | **73.2** | **66.4** | 95.2 | **73.2** | **66.4** | 95.2 | **73.2** | **66.4** | 95.2 |
| Improvement | +10.9 | +15.5 | -1.2 | +14.2 | +20.4 | -1.8 | +8.9 | +14.6 | -1.6 |
| Ours (PointNet++) | 73.2 | 66.4 | 95.2 | 73.2 | 66.4 | 95.2 | 73.2 | 66.4 | 95.2 |
| Improvement | +10.9 | +15.5 | -1.2 | +14.2 | +20.4 | -1.8 | +8.9 | +14.6 | -1.6 |
| Ours (DGCNN) | 73.2 | 66.4 | 95.2 | 73.2 | 66.4 | 95.2 | 73.2 | 66.4 | 95.2 |
| Improvement | +10.9 | +15.5 | -1.2 | +14.2 | +20.4 | -1.8 | +8.9 | +14.6 | -1.6 |

# D   Results on Large-Scale Datasets

The large-scale 3D datasets (MVimgNet [43]) are available and we conduct an MVimgNet50-100 task as shown in the Tab.11. In this task, we selected 50 classes as known classes and another 100 classes as novel classes from the large-scale MVimgNet dataset. Despite the challenges when increasing the number of unknown classes, our method can still outperform baselines by a large margin.

Table 11: The results on large-scale datasets.

|  | Novel | Known |
|---|---|---|
| AutoNovel [10] | 12.8 | 47.8 |
| NCL [48] | 11.3 | 53.9 |
| UNO [7] | 10.7 | 44.3 |
| IIC [19] | 12.5 | 58.2 |
| Ours | **43.9** | **64.0** |
| Improvement | +31.1 | +5.8 |

# E   Estimate Cluster Numbers

We fixed the number of unseen classes following the baseline methods [10]. However, the number of novel classes can be estimated by following the strategy in AutoNovel [10] and GCD [35], where semi-supervised k-means and a validation set can be used to select the optimal number of unseen classes. We estimate the number of clusters using the above method on the ModelNet similarity task as shown in Tab. 12. The results show the estimation method may slightly underestimate the novel class number when similarities between novel classes and known classes are high. This is because some highly related classes are clustered together. Meanwhile, it slightly overestimates the number of novel classes when similarities decrease.

Table 12: Estimate cluster numbers on modelnet dataset.

|  | High | Medium | Low |
|---|---|---|---|
| Ground Truth | 26 | 27 | 27 |
| Ours | 24 | 30 | 31 |

# F   Implementation Details

In all experiments, we adopt PointNet [24] as the backbone for point-wise feature learning and sample $1024$ points from the original point clouds as inputs. We use scaling, rotation, jittering, and translation as data augmentation to generate two augmented point clouds for both labeled and unlabeled inputs respectively during training. The network is optimized with the Adam optimizer with an initial learning rate of $0.001$. In our experiments, we first train the network for 30 epochs with labeled dataset $D^l$ to learn the part concept bank, then we train all networks for another $220$ epochs with both labeled and unlabeled datasets. We specify both the feature vector size $D$ and the dimension of the part concepts to $256$. The overall method is implemented in PyTorch, and we conduct all experiments on a computer with an NVIDIA Tesla V100 GPU. Our method takes 1.26M parameters, 14.7G memory cost and 0.43s train time per batch measured on ModelNet. Readers are encouraged to refer to the released code for detailed implementation.

