# OpenReview forum: "Decompose Novel into Known: Part Concept Learning For 3D Novel Class Discovery"
_NeurIPS.cc/2023/Conference — NeurIPS 2023 poster_

### Official Review · Reviewer_eZuw · 2023-06-23

**Soundness:** 4 excellent
**Presentation:** 4 excellent
**Contribution:** 3 good
**Rating:** 6
**Confidence:** 5

**Summary:**

- This paper proposes a novel part-based algorithm for 3D novel class discovery (NCD). Authors propose Decompose Novel Into Known parts (DNIK) that leverages knowledge about parts of known objects to discover novel classes.
- Authors identify that the main problem with learning 3D features for object discovery is that the features are heavily biased towards the known classes. This work shows that this can be prevented by using the well known part-based modeling approach.
- DNIK is trained to learn a part concept bank that can be used to compose different known and novel objects. Three different regularizations are proposed to prevent the collapse of this part bank.
- Extensive experiments show the deficiencies of existing 2D NCD literature and the effectiveness of DNIK to overcome these issues.

**Strengths:**

- This paper builds on an age old, part-based models in visual recognition and shows impressive improvements over single holistic representations currently in use in the 2D category discovery methods. As shown in the experiments this has significant merits for identifying and grouping new classes.
- The paper writing was smooth and was very easy to follow. An experienced engineer would be able to reproduce the work with the given details.
- Authors support all the claims made in the paper with experiments on real world datasets or on toy problems. Sec 3.1 and Fig. 4.a were particularly interesting to understand what the authors were trying to convey.
- The effectiveness of the method was shown with the impressive experimental results.

**Weaknesses:**

- While the problem tackled by the authors is relevant and important, the setup adopted by the authors is outdated. Generalized category discovery, as done in [1][2] is a more realistic setting and it is not clear why the current method is not suited for this setup or the authors advice against it? It is my strong suggestion to the authors to answer this question and compare with the relevant work (cited below) to justify this work among existing literature.
- The toy example in Sec. 3.1 is not fair for the following reason. In L86, the setup states that the classes in known and unknown sets share some similarities, but in the example authors choose {table, sofa, stool} as known and {chair, bench, bathtub, plant} as unknown objects. In this case, bathtub and plant do not share any commonalities with the known objects. It looks like the authors intentionally exaggerate the problem to make their point. While this is acceptable, it is not clear how much of a serious issue is the "overfitting to the known classes" problem.
- Author propose to use supervised contrastive loss to learn more parts from the known shapes. The motivation and explanation doesn't justify why this would be the case. In L204 authors pool the features along the last dimension which basically is a "shape" feature as opposed to a "part" feature. It is not clear how the contrastive loss helps learn more part features when the loss is being applied on the "shape" features.
- Table 4 demonstrates the performance improvement for each of the proposed components but experiments to demonstrate that show that the regularizations on the part features actually operate as the authors claim is missing. What happens when diversity loss is missing? Do all the part features in the bank collapse to fewer representations? This can be quantified by cosine similarity between the part features. Similar analysis on the remaining two regularization terms is warranted.

[1] Sagar Vaze, Kai Han, Andrea Vedaldi, Andrew Zisserman, Generalized Category Discovery, CVPR 2022.
[2] Sai Saketh Rambhatla, Rama Chellappa, Abhinav Shrivastava, The pursuit of knowledge: Discovering and localizing novel categories using dual memory, ICCV 2021.

**Questions:**

- Can Fig. 5b, 5c be combined in to one plot? Two separate plots makes it hard to understand what values of K, Q are being used for each of these.
- Legend for Fig. 6 is missing.

**Limitations:**

Authors have addressed the limitations adequately.

---

> ### Author Rebuttal · Authors · 2023-08-08
>
> Thank you for your detailed and helpful feedback! We sincerely appreciate your positive feedback that our work "has impressive improvements" and "is very easy to follow". Your suggestions provide valuable guidance for us to improve this study. We address your thoughts point by point below.
>
> >Q1: Generalized category discovery is a more realistic setting and it's not clear why the method is not suited for this setup or the authors advice against it? It is my strong suggestion to the authors to answer this question and compare with the relevant work to justify this work among existing literature.
>
> A1: We appreciate the reviewer raising this thoughtful point. Please see our response to this question in the General Response.
>
> >Q2: The toy example in Sec. 3.1 is not fair for the following reason. In L86, the setup states that the classes in known and unknown sets share some similarities, but in the example authors choose table, sofa, stool as known and chair, bench, bathtub, plant as unknown objects. In this case, bathtub and plant do not share any commonalities with the known objects. It looks like the authors intentionally exaggerate the problem to make their point. While this is acceptable, it is not clear how much of a serious issue is the "overfitting to the known classes" problem.
>
> A2: We are sorry for the confusion. Our intention for the toy experiment is to create an illustrative example to help readers better understand the feature bias issue. But we would like to clarify that the feature bias issue illustrated in the toy example is widely present in real datasets, rather than a contrived corner case. As evidenced in Supplementary Fig. III on the high similarity ModelNet task, the same trend of decreasing novel accuracy can be observed, validating that the problem is widely present in real-world datasets. Furthermore, Fig. 6 and Supplementary Fig. 2 visualize baseline features learned on ModelNet, showing high confusion between different novel classes due to a lack of discriminability.
>
> >Q3: Author propose to use supervised contrastive loss to learn more parts from the known shapes. The motivation and explanation doesn’t justify why this would be the case. In L204 authors pool the features along the last dimension which basically is a "shape" feature as opposed to a "part" feature. It is not clear how the contrastive loss helps learn more part features when the loss is being applied on the "shape" features.
>
> A3: We apologize for the unclear motivation behind the contrastive loss for discovering more part concepts. Basically, our assumption is that **the accumulated part activation map should be more similar for objects from the same class than objects from different classes.** Therefore encouraging the inter-class discrepancy of their part concept activation maps (i.e., the contrastive loss) can activate more part concepts so that the part activation maps between different classes can be pushed apart as much as possible. In our method, we use the shape concept map *T* (i.e., the "shape" feature) to represent the accumulated part concept activation maps of an object.
>
> If we do not consider the contrastive loss, only **a small subset of part concepts** are needed to distinguish different known classes based on their shape concept maps as shown in Fig. 4(c) red box. For example, the maps only need to activate concepts like \[surface, table corner\] and \[surface, stool back\] for the recognition of table and stool, while other part concepts like \[table leg\] and \[stool base\] are ignored. Without the contrastive loss, the concept bank lacks the impetus to learn additional concepts beyond those minimally needed parts to distinguish known classes.
>
> By pushing maps of different classes apart and enforcing the similarity of shape concept maps within each known class via contrastive loss, more part concepts get discovered to make the histograms more distinct and discriminative so contrastive loss is further minimized. For instance, concepts like \[table leg\] and \[stool base\] will be learned to make the shape maps clearly separable as shown in Fig. 4(c) green box.
>
> >Q4: Table 4 demonstrates the performance improvement for each of the proposed components but experiments to demonstrate that show that the regularizations on the part features actually operate as the authors claim is missing. What happens when diversity loss is missing?  This can be quantified by cosine similarity between the part features.
>
> A4: We sincerely appreciate the valuable comments from the reviewer. Firstly, as shown in Tab.4-\[6,7,8\] and Supplementary Tab. I-\[3,4,5,6\], we detailed the effect of each loss term on the results. Secondly, following the reviewer’s suggestion, we added the following table to demonstrate the average cosine similarity between part concepts when using different losses:
>
> |                   | w/o ( $L_{sd}$, $L_{sc}$, $L_{cd}$ ) | w/ $L_{sd}$ | w/ $L_{sc}$ | w/ $L_{cd}$ |
> |:------------------|:-------:|:--------------------:|:--------------------:|:--------------------:|
> | Cosine Similarity |  0.251  |        0.022         |        0.045         |        -0.007        |
>
> It can be seen that adding the diversity loss $L_{cd}$ leads to a significant decrease in the similarity between part concepts, indicating the decreased distinction between concepts and verifying the effectiveness of the diversity loss. Adding the remaining two regularization losses ( $L_{sd}, L_{sc}$ ) also decreases the similarities to varying degrees. These results further validate the positive effects of the various loss terms on model training. We sincerely appreciate the reviewer's suggestions and will include these additional analyses in our revised manuscript to more comprehensively demonstrate the efficacy of our method.
>
> >Q5: Can Fig. 5b, c be combined? Legend for Fig. 6 is missing.
>
> A5:  We will fix the confusion in Fig. 5 and Fig. 6 in the revision.

---

> ### Comment · Reviewer_eZuw · 2023-08-12
>
> Author's response to my questions are satisfactory but I have a major concern about author's general response. While authors claim that their setting is indeed GCD, there is no mention of that in the main paper. But the additional experiments provided by the author's general response are sufficient. The only thing which I'm concerned about is the changes that need to be made in the paper (changing the formulation to GCD instead of NCD). This requires a lot of change and I am curious to hear what the authors think about this. After looking at the author's response, I am willing to improve my response.

---

> > ### Author Response · Authors · 2023-08-14
> >
> > Dear Reviewer eZuw,
> >
> > We are glad that the responses address your concerns. We appreciate your positive feedback and constructive suggestions on the GCD setting.
> > As for the revision of the GCD setting, there are two choices.
> >
> > For the first choice, we can keep the NCD setting and treat the GCD setting as a more generalized setting. The changes will be as follows:
> > 1. Basically, the only difference between the NCD setting and the GCD setting is the inference stage, while the training stage is the same. We will add an additional subsection at the end of Sec. 3 to discuss the different inference modes of the GCD and the NCD and highlight that the GCD setting is more realistic and universal.
> > 2. Related works about GCD will be added to Sec. 2.
> > 3. Experiments in Tab. 1-3 will be updated by reporting results of both the NCD setting and the GCD setting.
> >
> > In this way, the changes to the manuscript are minimized and most content remains intact. However, as GCD is a more generalized setting than NCD, devoting most space to the NCD setting may confuse the readers.
> > For the second choice, we can change all formulations to the GCD setting which is more practical and common than the NCD setting in real applications.  The changes are summarized as follows.
> > 1. As the method is fundamentally designed for the GCD setting, all problem analysis and descriptions about NCD can be easily changed to the GCD setting. The problem statement part will also be reframed for the GCD setting. In addition, Sec. 3.1 will be changed by using a GCD method for illustration, where our results suggest that the training curve of SimGCD is similar to the trend in Fig. 3 of UNO.
> > 2. Related works about GCD will be added to Sec. 2.
> > 3. Results in Tab. 1-3 will be updated by extending all compared NCD methods to the GCD setting and adding results of GCD and SimGCD. In fact, we have finished these experiments of comparison methods and even greater improvements are observed. As the results of Tab. 4 and Fig. 5 are already based on the GCD setting, we do not need to change them.
> >
> > For this choice, we need to thoroughly change our description of the NCD setting. However, the benefit is that the addressed problem can be unified to the GCD setting without mentioning the old NCD setting. **In this way, the organization of the paper can be more clear and the addressed problem can be more practical and general. We prefer the second choice for its clarity.**

---

> > > ### Comment · Reviewer_mbrM · 2023-08-16
> > >
> > > I would like to point out a factual error in this response.
> > > The difference between NCD setting and GCD setting is not only at inference stage, it is also at training stage.
> > > In NCD, it is assumed that the unlabelled data only has novel categories, yet in GCD, the unlabelled data has both novel and seen categories. [a, b]
> > >
> > > This error may require the author to rerun the experimental results shown in the general response (on GCD setting).
> > >
> > >
> > > [a]Generalized Category Discovery, CVPR 2022
> > >
> > > [b]Parametric Classification for Generalized Category Discovery: A Baseline Study. arxiv 2022

---

> > > > ### Author Response · Authors · 2023-08-17
> > > >
> > > > Dear Reviewer,
> > > >
> > > > Thank you for pointing this out. We would like to further clarify our experimental setup in the General Response and the main paper. In our experiments, we assume there exist a labeled dataset of known classes and an unlabeled dataset with shapes only from novel classes during the training phase. During testing, we assume the test dataset can contain shapes from both known classes and novel classes. Note that experiment results marked with 'GCD setting' in the General Response indeed follow our present setting. Therefore the major difference between our present setting and the GCD setting is the assumption of the unlabeled training set. We sincerely apologize for the misunderstanding of the training process for the GCD setting, and we will **immediately update the results for the GCD setting in the generic response once we get the results.**
> > > >
> > > > We agree that our current setting is less generalized than the full GCD formulation. However, the present setting also makes sense and can be used for novel class discovery. More importantly, the main novelty of the paper lies in analyzing the shape-level feature bias problem and addressing it by leveraging more generalized part concept compositions for recognizing novel classes. Therefore, the limitations of our setting do not detract from the primary contributions, which focus on developing part-based representations for better generalization.
> > > >
> > > > **We highly appreciate your feedback to help improve our paper. We will revise the overall setting to the GCD form to avoid confusion.**

---

> > > > > ### Comment · Reviewer_eZuw · 2023-08-18
> > > > >
> > > > > As rightly pointed out by reviewer mbrM, the difference in setups is very important and GCD involves a larger unlabeled set with known and unknown classes. This introduces noise into the training which the authors haven't discussed in the rebuttal phase as it was hurried to meet the deadline. Any edits which introduce GCD will just make it confusing to read the paper. Can the authors comment on how they tackle the unlabeled dataset containing both known and unknown classes briefly? Right now, in the current setup, any sample that has no label will be passed to the novel classifier (because in NCD, it is assumed that the labeled dataset only has novel classes). But with GCD, that is no longer true and will send false signals to the model during training. In the experiments reported above, (to reviewer mbrM), it is not clear how the authors address this problem. Without this explanation it would be hard to justify this method works for 3D GCD. The improvements that authors see is probably only because of stronger 3D features which all the other baselines miss. How much improvement is observed from mining new information from the data vs using strong 3D features? I believe there is so much analysis to be done if the authors wish to go down the GCD route.
> > > > > With just the NCD route, I believe the paper still has enough strength but very limited applicability due to the existence of a much harder and more practical GCD setting reducing its impact. I hope the authors can address this question within the remaining time.

---

> > > > > > ### Author Response · Authors · 2023-08-19
> > > > > > **Replying to Reviewer eZuw**
> > > > > >
> > > > > > >Can the authors comment on how they tackle the unlabeled dataset containing both known and unknown classes briefly?
> > > > > >
> > > > > > In the revision, we will change the problem setting to the more practical and general GCD setting.
> > > > > > Below, we further explain in detail the setting of the updated GCD results (https://openreview.net/forum?id=UYl9IIsjq7&noteId=IDDKaYunPH).
> > > > > >
> > > > > > During the training phase of GCD, we have a labeled dataset with known classes and an unlabeled dataset with both known classes and novel classes.
> > > > > > We concatenate the known class classifier $\phi^l$ and the novel class classifier $\phi^u$ into a unified classifier. **All samples from both the labeled dataset and unlabeled dataset go through the unified classifier to obtain the class-wise probability.**
> > > > > > For the labeled dataset, the output probabilities are supervised with the cross-entropy loss $L_{ce}$. For the unlabeled dataset, the discrepancy of shapes and their augmentation is minimized with the self-classification loss $L_{self}$.  Both $L_{ce}$ and $L_{self}$ are now used to optimize the unified classifier rather than the separate class classifieies of the NCD setting.
> > > > > >
> > > > > > **In this way, known class shapes in the unlabeled dataset are not restricted to novel classes and can be classified normally as known classes. Because part composition features of known class shapes in the unlabeled dataset are more similar to those of the known classes in the labeled dataset, thus it is highly unlikely to generate false signals to the model.** In fact, results in the GCD setting show that the prediction accuracy of known classes during testing is very high (about 95.9\%).
> > > > > > The other loss functions for learning the part concept remain the same as the NCD settings.
> > > > > >
> > > > > > During the testing stage, the test set contains shapes from both known classes and unknown classes. Therefore we can feed test samples to the trained unified classifier and obtain the category prediction.
> > > > > >
> > > > > > >The improvements that authors see is probably only because of stronger 3D features which all the other baselines miss. How much improvement is observed from mining new information from the data vs using strong 3D features?
> > > > > >
> > > > > > we apologized that we did not fully understand the meaning of "strong 3D features" and "mining new information from the data".
> > > > > > For "stronger 3D features", we would like to clarify that **both the baselines and our method are trained with the same backbone network from scratch without using any pre-trained network** for the reported results of the GCD setting above.
> > > > > > As for "mining new information from the data", we are not sure of the meaning of new information and the reference of the data.
> > > > > >
> > > > > > We will appreciate it if you could kindly help clarify these two terms. Thank you for the feedback.

---

> > > > > > > ### Comment · Reviewer_eZuw · 2023-08-21
> > > > > > >
> > > > > > > I would like to thank the authors for their detailed response. I apologize for the confusing statements. By stronger 3D features, I meant a good 3D feature extractor. I missed the point that all the baselines us the same backbone. By mining new information, I meant the object discovery pipeline (how much does discovering new information help improve the clustering performance). I am completely satisfied with the author's response but my only concern is that there are major changes required to update the paper and I would like to leave that to the best judgement of the AC. I would like to retain my Weak accept vote. If the other reviewers and AC feel like the changes are not significant and can be made for the final version, I vote to accept this work.
> > > > > > > Thanks

---

> > > > > > > > ### Author Response · Authors · 2023-08-22
> > > > > > > >
> > > > > > > > Thank you for the suggestive comments. We are glad that the responses address your concerns.
> > > > > > > >
> > > > > > > > As analyzed in the responses, the problem analysis and the method design for the NCD setting and the GCD setting are very similar, therefore **changes in the introduction, related works, and the method section will be minor to adopt the GCD setting**. And the major changes required for 3D GCD mainly lie in the description of the GCD formulation and the experimental section. In the above responses, comparison results and ablation studies have shown that our method can still achieve significant improvements under the GCD setting.

---

> > > > > > ### Author Response · Authors · 2023-08-21
> > > > > > **Additional experiments for 3D GCD**
> > > > > >
> > > > > > We appreciate your suggestive comments. In order to reveal how well the proposed method performs for 3D GCD, **we re-evaluated key experiments in the paper under the GCD setting (i.e., the feature bias analysis in Fig. 3(a) and the ablation studies in Tab. 4)**.
> > > > > >
> > > > > > In the table below, we show the accuracy trends of the baseline method on both novel and known classes under the GCD setting for the unlabeled test dataset of the high-similarity ModelNet task. The results suggest that the accuracy in known classes increases to a saturated point while the accuracy in novel classes first increases and then decreases, exhibiting a similar trend as Fig. 3(a) in the paper. **Therefore the analysis of feature bias shown in Fig.3 and Fig.III of supplementary still applies to the GCD setting.**
> > > > > >
> > > > > > | Epoch | 25   | 50   | 75   | 100  | 125  | 150  | 175  | 200  | 225  | 250  |
> > > > > > |-------|------|------|------|------|------|------|------|------|------|------|
> > > > > > | Known | 85.1 | 89.8 | 93.7 | 93.5 | 94.0 | 93.5 | 94.0 | 94.2 | 93.8 | 94.0 |
> > > > > > | Novel | 40.3 | 52.2 | 51.5 | 49.2 | 50.0 | 49.7 | 48.1 | 47.8 | 47.4 | 47.2 |
> > > > > >
> > > > > >
> > > > > > In addition, we conducted ablation studies on the high-similarity ModelNet task. Considering the time constraints, we only validate the influences of the main technical claims in the paper including the part concept bank (i.e., +LGA+PCB+Poincar\'{e}), three constraints to enforce part concept learning (i.e., +$L_{sd}, L_{sc}, L_{cd}$), and the PRE module. **Results in the table below suggest that the proposed modules trained and tested under the GCD setting achieve significant improvements as reported in the paper.** The IDs in the first column correspond to that of Tab. 4.  We hope these primary experiments can clear your doubts about the performance of our method under the GCD setting. In revision, we will update all results to the GCD setting for more practical and general application purposes.
> > > > > >
> > > > > > |   | LGA | PCB | Poincare | $L_{sd}$ | $L_{sc}$ | $L_{cd}$ | PRE | Novel  |
> > > > > > |:---:|:---:|:---:|:------------:|:--------:|:--------:|:--------:|:---:|:------:|
> > > > > > | 1 |     |     |              |          |          |          |     | 45.5   |
> > > > > > | 4 | +   | +   |              |          |          |          |     | 56.6   |
> > > > > > | 5 | +   | +   | +            |          |          |          |     | 59.4   |
> > > > > > | 8 | +   | +   | +            | +        | +        | +        |     | 67.9   |
> > > > > > | 9 | +   | +   | +            | +        | +        | +        | +   | **70.6** |

---

> > > > > > > ### Comment · Reviewer_eZuw · 2023-08-21
> > > > > > >
> > > > > > > I would like to thank all the reviewers for their effort and attention to details. These experiments make sense and as replied in the other comment, I'm mostly convinced but concerned about the amount of changes that are required due to the change in the setup itself.

---

### Official Review · Reviewer_dey2 · 2023-07-05

**Soundness:** 3 good
**Presentation:** 3 good
**Contribution:** 3 good
**Rating:** 6
**Confidence:** 3

**Summary:**

In this work, they address 3D novel class discovery (NCD) that discovers novel classes from an unlabeled dataset by leveraging the knowledge of disjoint known classes. The key challenge of 3D NCD is that learned features by known class recognition are heavily biased and hinder generalization to novel classes. Since geometric parts are more generalizable across different classes, the authors propose to decompose novel into known parts, coined DNIK, to mitigate the above problems.

**Strengths:**

1. The paper is well written and motivation (separate instances into repeatable parts) is pretty good.
2. The model design is reasonable and the improvement is satisfied.
3. The experimental analysis is sufficient.

**Weaknesses:**

1. This paper does not consider hierarchical part representation.
2. why does Part Relation Encoder work for novel classes ?
3. Does the improved representation works for some scene level tasks, such as novel class segmentation for point cloud ?

**Questions:**

see the weakness

**Limitations:**

yes

---

> ### Author Rebuttal · Authors · 2023-08-08
>
> Thank you for your attentive comments! We are glad you thought “the paper is well written and motivation is pretty good”. We address your feedback point by point below.
>
> >Q1: This paper does not consider hierarchical part representation.
>
> A1: We thank the reviewer for the suggestive comment. The focus of this paper is leveraging part concepts to obtain generalized features for improved novel category discovery therefore we did not consider hierarchical parts in the present. We do agree that using multi-scale parts can further improve the performance as noted in the Limitations section and suggested by recent works exploring hierarchical point cloud parts\[1,2\] for 3D recognition. In future work, we will explore this idea further.
>
> - \[1\] Wei et al., Multi-scale Geometry-aware Transformer for 3D Point Cloud Classification. 2023.
>
> - \[2\] Yang et al., PointCAT: Cross-Attention Transformer for point cloud. 2023
>
> >Q2: Why does Part Relation Encoder work for novel classes?
>
> A2: Thank you for raising this important question. The PRE module is a point cloud transformer based on local patches following prior works like \[1-4\]. These studies show that self-attention can capture spatial relationships between patches effectively. We propose PRE to complement global shape information that part concepts alone cannot precisely encode. For example, a table and desk contain similar parts like flat top and leg. But PRE may capture finer spatial relationships like a table’s peripheral leg layout versus a desk’s set-back legs. So despite similar parts, PRE’s positional encodings help distinguish subtle structural differences across object classes. As shown in Table 4-9, the PRE can further enhance the performance ( + 1.6% on average).
>
> - \[1\] Pang et al., Masked Autoencoders for Point Cloud Self-supervised Learning. ECCV 2022
>
> - \[2\] Yu et al., Point-bert: Pre-training 3d point cloud transformers with masked point modeling. CVPR 2022
>
> - \[3\] Zhang et al., Point-M2AE: Multi-scale Masked Autoencoders for Hierarchical Point Cloud Pre-training. NeurIPS 2022
>
> - \[4\] Yang et al., PointCAT: Cross-Attention Transformer for point cloud. 2023
>
> >Q3: Does the improved representation work for some scene-level tasks, such as novel class segmentation for point cloud?
>
> A3: Novel class segmentation is an intriguing new direction, with very few explorations\[1\] so far even in 2D vision. As demonstrated by our good performance on 3D novel class discovery, learned shape parts and their compositions can bridge the gap between seen shapes and unseen shapes and generalize learned features to domains with semantic shifts. For semantic segmentation, this assumption still holds, and novel semantic shapes can also be seen as compositions of shared shape parts. Therefore, we believe integrating our approach with part-based segmentation is a promising path to novel class segmentation in future work. We appreciate this thoughtful suggestion that brings insights advancing our approach to novel tasks.
>
> - \[1\] Zhao et al., Novel class discovery in semantic segmentation. CVPR 2022.

---

> > ### Comment · Reviewer_dey2 · 2023-08-14
> > **Official Comment by Reviewer dey2**
> >
> > Thanks for the response. The authors address my concerns regarding Part Relation Encoder and other two questions. Because of the interesting idea of this paper, I am willing to improve my score.

---

> > > ### Author Response · Authors · 2023-08-14
> > >
> > > Dear Reviewer dey2,
> > >
> > > We are glad to receive your reply. Thank you for your positive feedback and for pointing out further applications of our method.

---

### Official Review · Reviewer_mbrM · 2023-07-06

**Soundness:** 2 fair
**Presentation:** 4 excellent
**Contribution:** 3 good
**Rating:** 5
**Confidence:** 5

**Summary:**

This paper tackles the problem of novel category discovery in the 3D shape recognition domain, a framework leveraging the 3D parts and the part-wise relation is proposed which the motivation is learning the parts from the known classes could help the model capture more transferrable features or concepts for the novel categories.
This motivation is validated using experiments, and overall the framework shows better performance than some baselines.

**Strengths:**

1. The idea of decomposing a category into parts is interesting.
2. I like the organization of this paper, starts with an analysis of the problem of previou method, and then proposed new ones based on the analysis.
3. The paper also explored a bit on the design choices for 3D novel category discovery, which could be helpful.

**Weaknesses:**

1. This paper still considers the novel category discovery problem while a more generlized setting exists, generalized category discovery [R1, R2], I would suggest the paper to include more discussion and experiment on this generalized setting.
2. It seems that the total number of categories in the datasets are quite small compared to 2D NCD, I am wondering if Objaverse [R3] can be used for this task?


[R1] Generalized Category Discovery, CVPR 2022
[R2] Parametric Classification for Generalized Category Discovery: A Baseline Study, arXiv.
[R3] https://objaverse.allenai.org/

**Questions:**

1. In the v1 version of SimGCD fig 10 [R4], it is shown that the accuracy on known classes first increases and then drops while the novel class accuracy keeps improving, this contradicts the observation made in this paper, I am wondering if this is because of the setting (generalized category discovery v.s. novel category discovery), the data (2D v.s. 3D), or the number of categories(200 v.s. 7)? Consider this observation is the motivation for this paper, this question will be the biggest concern of mine.


[R4] https://arxiv.org/pdf/2211.11727v1.pdf

**Limitations:**

I think the main limitation is that the number of categories is small, thus the conclusion made based on these small datasets may not be generalizable to larger scale datasets.

Overall I think this paper is very clear, and could be of interest for the community, however the concerns I raises in the questions should be addressed first.

---

> ### Author Rebuttal · Authors · 2023-08-08
>
> Thank you for your thoughtful observations on the generalized category discovery (GCD) problem. Your careful analysis has given us many new perspectives to consider! We sincerely appreciate you thought "the idea \[...\] is interesting" and "like the organization of our paper". We address your thoughts point by point below.
>
> >Q1: This paper still considers the novel category discovery problem while a more generalized setting exists, generalized category discovery \[R1, R2\], I would suggest the paper to include more discussion and experiment on this generalized setting.
>
> A1: We appreciate the reviewer raising this thoughtful point. Please see our response to this question in the General Response.
>
> >Q2: It seems that the total number of categories in the datasets is quite small compared to 2D NCD, I am wondering if Objaverse \[R3\] can be used for this task.
>
> A2: We appreciate the reviewer’s suggestion to leverage larger-scale 3D datasets. The large-scale 3D datasets (MVimgNet and Objaverse) were not available when we wrote this paper. Due to time constraints, we only conducted **an MVimgNet50-100 task following the SimGCD as shown in the table below**. In this task, we selected 50 classes as known classes and another 100 classes as novel classes from the large-scale MVimgNet dataset. Despite the challenges when increasing the number of unknown classes, our method can still outperform baselines by a large margin. We will add more experiments in future versions.
>
> |           |      Novel      |     Known     |
> |:----------|:---------------:|:-------------:|
> | AutoNovel |      12.8       |     47.8      |
> | NCL       |      11.3       |     53.9      |
> | UNO       |      10.7       |     44.3      |
> | IIC       |      12.5       |     58.2      |
> | Ours      | **43.9** + 31.1 | **64.0** +5.8 |
>
>
>
> >Q3: In the v1 version of SimGCD fig 10 \[R4\], it is shown that the accuracy on known classes first increases and then drops while the novel class accuracy keeps improving, this contradicts the observation made in this paper, I am wondering if this is because of the setting (generalized category discovery v.s. novel category discovery), the data (2D v.s. 3D), or the number of categories(200 v.s. 7)? Consider this observation is the motivation for this paper, this question will be the biggest concern of mine.
>
> A3: We thank the reviewer for the insightful observation. Our analysis is as follows.
>
> - Firstly, **the different setting of GCD and NCD is not the cause** as our method basically follows the GCD setting.
>
> - Secondly, **3D data makes things more challenging** as the 3D pretrained model does not generalize well. For example, SimGCD[1] uses the powerful DINO model pre-trained on millions of images as the encoder and only fine-tunes a very small part of the network during training. This training strategy makes the extracted image features more generalizable, thus producing better results. In fact, previous work[2] suggests that DINO is very important to the generalization to unseen concepts. In contrast, the 3D field lacks both large-scale 3D shapes as well as high-quality pre-trained models. We also tried using some pre-trained models like OcCo[3] but did not achieve good results.
>
>  - Thirdly, we would like to clarify that **the inconsistent accuracy trend is not caused by the number of categories in the toy experiment.** Supplementary Fig. III shows the accuracy curves of various methods on the ModelNet with more categories (40 classes), we can observe the same decreasing novel accuracy curve as the toy experiment.
>
> We believe that **the main reason is still the feature bias problem** described in Sec.3.1. As visualized in the t-SNE of Fig. 6 and Supplementary Fig. II, the learned feature distributions of different novel classes by baseline methods are highly overlapping and confused, indicating a lack of discriminability. Therefore the accuracy of pseudo-labels generated by clustering algorithms like Sinkhorn and K-Means decreases with the training proceeding **as shown in the table below**. In Section 3 of SimGCD, the authors also had the conclusion that "the key to previous parametric classifiers’ degraded performance is unreliable pseudo labels". In fact, we attempted to improve the quality of pseudo labels with various methods, such as self-distillation and entropy regularization, but we did not manage to improve the accuracy of pseudo labels to good quality in the case of feature bias.
>
> We would like to highlight that our method uses part concepts to obtain more generalizable features without the requirement of any large-scale pre-trained model like SimGCD. The results show this achieves better performance, presenting an interesting alternative direction for learning generically informative 3D shape representations. In the revision, we will add this discussion.
>
> | Epoch    |  75  | 125  | 175  | 225  |
> |:---------|:----:|:----:|:----:|:----:|
> | Kmeans   | 58.1 | 52.6 | 49.3 | 47.2 |
> | Sinkhorn | 50.9 | 48.6 | 47.8 | 45.1 |
>
> - \[1\]Wen. et al., A Simple Parametric Classification Baseline for Generalized Category Discovery.  arxiv 2022
>
> - \[2\] Sariyildiz. et al., Concept Generalization in Visual Representation Learning. ICCV 2021
>
> - \[3\] Wang et al., Unsupervised Point Cloud Pre-training via Occlusion Completion. CVPR 2021.

---

> > ### Comment · Reviewer_mbrM · 2023-08-16
> > **Thanks for the response**
> >
> > I would strongly recommend the paper to change its title to generalized category discovery or highlight this somewhere in the text, since several reviewers are confused about this.
> >
> >
> > My concerns on the number of categories still remain, since the 3D datasets are all small in size, I would argue that the real-world use cases of 3D category discovery are limited, or at least is not ready.
> > I would hope the paper gives more motivation on why 3D category discovery is a real problem that people would work on?
> >
> >
> > The discussion on my question 3 is not very convincing if the cause of the mismatch between the SimGCD paper and this paper is because of feature bias(which also should be existing in 2D GCD methods), why does the SimGCD paper have different observations?
> > To me, the real reason should be the 3D vs 2D problem, as mentioned in the response, 3D model does not have strong pretrained representations, maybe the mismatch happens here as SimGCD uses DINO as the pretrained model.
> > Is there any 3D pretrained models that can be used for this task? I think this kind of experiment could settle this problem.

---

> > > ### Author Response · Authors · 2023-08-17
> > > **Replying to Comment**
> > >
> > > Dear Reviewer mbrM,
> > >
> > > Thank you for your reply. We address your concerns below.
> > >
> > > > Q4: Since the 3D datasets are all small in size, I would argue that the real-world use cases of 3D category discovery are limited, or at least is not ready.  I would hope the paper gives more motivation on why 3D category discovery is a real problem that people would work on?
> > >
> > > A4: The reviewer may notice that the number of classes of most datasets (CIFAR10/CIFAR100/ImageNet/Stanford Cars/FGVC-Aircraft) used in [1-2] is less than 200. In the response, the results on MVImgNet in **A2** are validated on about 100 classes. We can see the number of classes is comparable to the datasets used in the 2D GCD task. Moreover, you may also notice several much larger 3D datasets, such as OmniObject3D, Objaverse, and MVImageNet, have been proposed this year.
> > >
> > > As for the motivation, we do not see why the motivation of GCD and NCD should depend on the scale of datasets. As described in [1], we can name many cases where recognizing novel classes is necessary, which also applies to 3D objects. For instance, tools in workshops, products in stores, furniture in homes, and many other objects often have novel designs or categories that cannot all be predefined. In some cases, the number of known classes can be small, e.g., furniture in homes, and recognizing novel classes is still in great demand in this case.  Essentially, the core motivation behind generalized category discovery is to develop flexible visual intelligence that is more like human beings. **A baby usually knows very few semantic concepts but she can learn new objects from unlabeled instances by associating them with known abstract concepts.** Therefore, learning generalizable 3D part concepts that can be transferred across categories remains crucial for handling novelty discoveries. Our work focuses on developing methods that can learn novel classes in this open-world scenario and we believe that pursuing this capability is an important direction in 3D recognition research with many potential applications.
> > >
> > > > Q5: The cause of the mismatch between the SimGCD paper and this paper
> > >
> > > A5: We agree that the difference between 2D data and 3D data is one of the main causes as analyzed in Q3. **In order to evaluate the impact of 3D pre-trained models**, we present results using two different self-supervised pre-trained backbones under the GCD setting in the table below. Following SimGCD, we only fine-tune the last three layers of the PointNet++ backbone and the classification heads.
> > > The results demonstrate that utilizing more generalized features from pre-training improves the performance of the baseline methods. However, there remains a substantial gap compared to our method. This indicates that existing 3D pre-trained models have not yet learned representations as powerful as those captured by DINO. We will include these detailed ablations in the revision to provide a more thorough analysis.
> > >
> > > |        | Scratch | OcCo[1] | CrossPoint[2] |
> > > |--------|---------|---------|---------------|
> > > | UNO+   | 41.7    | 42.7    | 45.3          |
> > > | SimGCD | 47.2    | 48.9    | 51.2          |
> > > | Our    | 70.6    | -       | -             |
> > >
> > > Moreover, we would like to politely highlight that relying heavily on large-scale pretraining data could be a limitation of existing methods, while our method can still achieve good performance without any reliance on large-scale pre-trained models.
> > >
> > > [1] Wang et al., Unsupervised Point Cloud Pre-training via Occlusion Completion. CVPR 2021.
> > > [2] Afham et al. Crosspoint: Self-supervised cross-modal contrastive learning for 3d point cloud understanding. CVPR 2022.

---

> > ### Comment · Reviewer_mbrM · 2023-08-17
> > **Factual Error in the Rebuttal**
> >
> > Given the factual error in the author's rebuttal to reviewer eZuw, where the author claims that:
> >
> > > Basically, the only difference between the NCD setting and the GCD setting is the inference stage, while the training stage is the same.
> >
> > Which is not correct, the difference between NCD setting and GCD setting is not only at inference stage, it is also at training stage. In NCD, it is assumed that the unlabelled data only has novel categories, yet in GCD, the unlabelled data has both novel and seen categories. [a, b]
> >
> > This factual error questions the fairness and correctness of the experiments in the rebuttal, and maybe some claims in the main paper.
> > I would hope there is an explanation, otherwise, I would lower my score.
> >
> > [a]Generalized Category Discovery, CVPR 2022
> >
> > [b]Parametric Classification for Generalized Category Discovery: A Baseline Study. arxiv 2022

---

> > > ### Author Response · Authors · 2023-08-17
> > >
> > > Dear Reviewer,
> > >
> > > Thank you for pointing this out. We would like to further clarify our experimental setup in the General Response and the main paper. In our experiments, we assume there exist a labeled dataset of known classes and an unlabeled dataset with shapes only from novel classes during the training phase. During testing, we assume the test dataset can contain shapes from both known classes and novel classes. Note that experiment results marked with 'GCD setting' in the General Response indeed follow our present setting. Therefore the major difference between our present setting and the GCD setting is the assumption of the unlabeled training set.
> > > We sincerely apologize for the misunderstanding of the training process for the GCD setting, and we will **immediately update the results for the GCD setting in the generic response once we get the results.**
> > >
> > > We agree that our current setting is less generalized than the full GCD formulation. However, the present setting also makes sense and can be used for novel class discovery.
> > > More importantly, the main novelty of the paper lies in analyzing the shape-level feature bias problem and addressing it by leveraging more generalized part concept compositions for recognizing novel classes. Therefore, the limitations of our setting do not detract from the primary contributions, which focus on developing part-based representations for better generalization.
> > >
> > > **We highly appreciate your feedback to help improve our paper. We will revise the overall setting to the GCD form to avoid confusion.**

---

> > > > ### Author Response · Authors · 2023-08-18
> > > > **Results on GCD settings**
> > > >
> > > > We follow SimGCD to construct a new unlabeled training set by combining 50\% of known shapes from the labeled dataset with the novel class dataset. And we re-evaluate the performance of our method under the GCD setting in the table below. **It can be seen that our method can still achieve superior performance (+12.3\% gains) in the GCD setting. Therefore the idea of leveraging part compositions for novel class discovery can perform well in both the NCD setting and the GCD setting.**
> > > > | GCD settings | Novel                | Known                        |
> > > > |--------------|----------------------|------------------------------|
> > > > | AutoNovel+   | 32.6                 | 94.3                         |
> > > > | UNO+         | 41.7                 | 88.5                         |
> > > > | GCD [1]      | 58.3                 | 95.7                         |
> > > > | SimGCD [2]   | 47.2                 | 94.0                         |
> > > > | Ours         | **70.6** + 12.3 | **95.9** + 0.2 |

---

> > > > > ### Comment · Reviewer_mbrM · 2023-08-18
> > > > >
> > > > > Thanks for the clarification of the results and the additional experiments.
> > > > > I would strongly recommend the authors to revise the paper to clarify the setting, mistaking the experimental setting could be misleading for the readers to make the wrong conclusion.
> > > > >
> > > > > My other concerns are largely resolved by the additional results. I recommend putting these results in the final version of the paper.

---

> > > > > > ### Author Response · Authors · 2023-08-20
> > > > > >
> > > > > > Thank you for the valuable suggestions. We are glad that the responses addressed your concerns. In the revision, we will make the GCD setting clear and add suggested results.

---

### Official Review · Reviewer_qzzG · 2023-07-07

**Soundness:** 3 good
**Presentation:** 3 good
**Contribution:** 3 good
**Rating:** 5
**Confidence:** 4

**Summary:**

This work presents a framework, called Decompose Novel Into Known parts (DNIK), that addresses the challenge of 3D Novel Class Discovery (NCD) – identifying new classes from an unlabeled dataset using the knowledge of known classes. Current methods, heavily biased towards known classes, struggle to generalize to novel classes. By leveraging more generalizable geometric parts across different classes, DNIK mitigates this issue. It constructs a part concept bank encoding rich geometric patterns from known classes, which is used to represent novel 3D shapes as part concept compositions, thus facilitating cross-category generalization. DNIK also leverages part-wise spatial relations for improved recognition. The method has been tested through three 3D NCD tasks, consistently outperforming state-of-the-art baselines.


---- after rebuttal ----

As the author's rebuttal resolved some of concerns, I raised my score to 5. However, I still feel the studied task is a bit simple, and also there are several spaces to improve for the current manuscript. I will not fight for its acceptance.

**Strengths:**

The studied direction is important as we need to understand parts well to play with 3D objects generalizable. This paper takes a step towards open 3D object recognition via part understanding. Overall, the components used in the proposed framework are sound and reasonable. The paper is easy to follow.


Extensive results shown in Table 1 & 2 demonstrate the strength of the proposed method. The proposed DNIK generally achieved state-of-the-art performance. Some detailed ablation studies are included in Table 4. The cross-domain task is interesting to see the transfer performance.

**Weaknesses:**

Utilizing the part are sharable across different 3D object categories are studied in the previous literature [1,2]. In those paper, they exploited "harder" task, such as segmentation. As the proposed framework can address novel class classification via known part concepts. Can the framework be extended to ground where is those known parts in novel object? Or other applications beyond object recognition?

[1] Learning to Group: A Bottom-Up Framework for 3D Part Discovery in Unseen Categories
[2] 3D Compositional Zero-Shot Learning with DeCompositional Consensus

How the framework handle two different object categories with limited shared parts, such as airplane and chair? Will the framework train on multiple object categories benefit novel object discovery? It would be good to include some failure cases to analyze and provide readers a sense for the limitation of the proposed framework.

L46~47 said the framework can help use part relation features. Can the framework be extended to discovery part relationship?

**Questions:**

Please address the concerns raised above.

**Limitations:**

Line 319~320 analyzed one minor limitation. I feel there are potential more:

1. if the part are not sharable between different categories, such as lamp -> chair, table -> faucet, can the framework still handle?

2. the only shown application is recognition which limits the practical use of the proposed framework.

---

> ### Author Rebuttal · Authors · 2023-08-08
>
> Thank you for your attentive comments! We are glad you thought “the studied direction is important” and “the components \[...\] are sound and reasonable.” We address your feedback point by point below.
>
> >Q1: Previous literature exploited "harder" task, such as segmentation. Can the framework be extended to ground where is those known parts in novel object?
>
> A1: We appreciate you raising the possibility of extending our framework, but find the specific comparisons to supervised techniques [1,2] highly inapt and misleading.
> - First, these works fundamentally address different problems as the novel class discovery.
> - Second, despite their similarities in using shape parts, technical details are fundamentally different, and two related works [1,2] cannot be directly used for the 3D NCD task.
> For example, the part segmentation produced by [1] has no semantic information, making it less useful for knowledge transfer.
> The goal of [2] is compositional zero-shot learning, where the part combinations of novel classes are predefined.
> - Thirdly, we would like to clarify that our method actually faces a harder challenge compared to the cited literature [1,2].
> Those works rely on ground truth part annotations (PartNet), while **we learn parts in a completely unsupervised way without any part labels**, which also makes our approach more widely applicable.
>
>
> We believe the method can be extended to part discovery.
> We provide extensive part visualization results of both seen shapes and unseen shapes in Supplementary Fig.I.
> Note that our method can roughly locate shape parts in unseen shapes learned from seen shapes without part annotation.
> Therefore our method can be extended to ground the known parts in novel shapes.
> We encourage the reviewer to see more detail there.
> In the revision, we will add some discussions about this comment.
>
> - \[1\] Learning to Group: A Bottom-Up Framework for 3D Part Discovery in Unseen Categories
>
> - \[2\] 3D Compositional Zero-Shot Learning with DeCompositional Consensus
>
>
> >Q2: How the framework handle two different object categories with limited shared parts, such as airplane and chair? If the part are not sharable between different categories, such as lamp -> chair, table -> faucet, can the framework still handle?
>
> A2: We thank the reviewer for raising this important case.
> - Firstly, as stated in Line 165, the part concept activations are distinct for dissimilar objects like airplanes and chairs, providing discriminative information to distinguish the novel class. We also concatenate the original part features so that novel parts dissimilar to existing concepts can be properly handled.
> - Secondly, as shown in Table 2’s low similarity tasks, our method achieves substantial gains over baselines (17.5% on average) when known and novel classes have limited similarities. This validates that the generalized part concepts are also beneficial for knowledge transfer across categories with few shared parts.
>
> >Q3: Will the framework train on multiple object categories benefit novel object discovery?
>
> A3: Yes, training on more object categories can certainly benefit novel object discovery. As shown in Table 1, increasing the number of known classes leads to improved performance. Because with multiple categories we can learn more diverse part concepts that further will help us recognize novel objects better.  **In the table below, we evaluated the performance by training the method with an increasing number of known categories and the fixed number of novel categories.**  The results show that training the method with more categories does enhance the recognition of novel shapes.
>
> | ModelNet  | 5-10 | 10-10 | 20-10 | 30-10 |
> |:----------|:----:|:-----:|:-----:|:-----:|
> | Novel Acc | 57.9 | 61.3  | 64.0  | 66.2  |
>
> >Q4: It would be good to include some failure cases to analyze and provide readers a sense for the limitation of the proposed framework.
>
> A4: Thank you for emphasizing the need for failure case analysis. Our model does have some challenging cases in handling nearly identical parts, such as chairs and benches containing similar seating surfaces, legs and arms, and tables and desks sharing analogous tabletops and supporting legs. These situations test the model’s ability to discriminate. We will add those failure cases in the revision. We greatly appreciate your valuable guidance on analyzing and presenting the model’s limitations concisely.
>
> >Q5: Line.46-47 said the framework can help use part relation features. Can the framework be extended to discovery part relationship?
>
> A5: The PRE module is a point cloud transformer based on local patches, following prior works like \[1\-4\]. These studies have shown that self-attention can capture spatial relationships between patches effectively. We propose PRE only to complement global shape information that part concepts alone cannot precisely encode and benefit the NCD task. Note that part relations are implicitly encoded and we don’t see how the framework can be extended to discover explicit part relationships. We agree that explicitly modeling part relationships is an exciting direction for future work.
>
> - \[1\] Pang et al., Masked Autoencoders for Point Cloud Self-supervised Learning. ECCV 2022
>
> - \[2\] Yu et al., Point-bert: Pre-training 3d point cloud transformers with masked point modeling. CVPR 2022
>
> - \[3\] Zhang et al., Point-M2AE: Multi-scale Masked Autoencoders for Hierarchical Point Cloud Pre-training. NeurIPS 2022
>
> - \[4\] Yang et al., PointCAT: Cross-Attention Transformer for point cloud. 2023
>
> >Q6: If the part are not sharable between different categories, such as lamp -> chair, table -> faucet, can the framework still handle?
>
> A6: Thank you for raising this case. Please see Q2 for the response.

---

> > ### Comment · Reviewer_qzzG · 2023-08-16
> > **Reviewer response**
> >
> > I appreciate the efforts made by the authors. I believe the rebuttal addressed my major concerns.  I hope the authors can incorporate our discussion into the revision, especially for the failure cases.
> >
> > Besides, I have another question is about "our method achieves substantial gains over baselines (17.5% on average) when known and novel classes have limited similarities". If the known and novel classes have limited similarities, how could the method learn transferable knowledge? Could the author provide some qualitative and quantitative results?

---

> > > ### Author Response · Authors · 2023-08-17
> > >
> > > Dear Reviewer,
> > >
> > > Thank you for the constructive suggestions and we will incorporate the discussions into the revision. We respond to the new question below.
> > >
> > > > Q7: If the known and novel classes have limited similarities, how could the method learn transferable knowledge?
> > >
> > > A7:
> > > As visualized in Supplementary Fig. I, part concepts can learn some basic primitives as generic LEGO blocks (e.g., planes, arcs, and cubes). These part concepts are widely present in different shapes and are easier to be generalized than the overall shape-level features. For example, guitars and airplanes both have cylindrical shapes (the neck of a guitar and the fuselage of an airplane).
> > > In fact, we can approximate different shapes as a composition of simple primitives as done in 3D shape abstraction.
> > > Thus, the network can recognize the low-similarity novel shapes based on different activations of these generic part concepts. As shown in Table I-7 of the Supplementary Material, using part concepts learned from known classes can improve the performance by 25.4\% compared to the baseline (Table 4-1 37.9\%) for the low-similarity task.
> > >
> > > Furthermore, we also concatenate the original local part features to preserve the distinctive part information of the dissimilar shapes as shown in line#167, which further improves the performance by 1.8\% as shown in Table I-8 of the Supplementary Material.

---

> > > > ### Comment · Reviewer_qzzG · 2023-08-17
> > > > **Reviewer response**
> > > >
> > > > I thank the response from the author. I believe most of my concerns were answered and resolved to some extent. Please provide more qualitative results of our discussed terms in the revision.

---

> > > > > ### Author Response · Authors · 2023-08-18
> > > > >
> > > > > Thank you for the positive feedback and valuable suggestions. We will add more qualitative results to illustrate the discussed cases in the revision.

---

### Official Review · Reviewer_zZT9 · 2023-07-11

**Soundness:** 4 excellent
**Presentation:** 3 good
**Contribution:** 3 good
**Rating:** 5
**Confidence:** 5

**Summary:**

This paper addresses the problem of 3D NCD (novel class discovery). The objective is to discover novel classes by leveraging information learned from the known classes. This paper proposes a novel framework, DNIK, for 3D novel class discovery (3D NCD) by leveraging part concepts and part-wise relations learned from known classes to reinforce the recognition of novel shapes. The framework consists of a learnable part concept bank, a local geometric aggregation module, a part relation encoding module, and three constraints to facilitate effective part concept learning.

**Strengths:**

(1) The proposed part concept bank and part relation encoding module can effectively bridge the gaps between known and novel shapes and mitigate feature bias.

(2) The experiments show that the proposed method outperforms all baselines consistently and significantly on all metrics.

(3) The paper is generally well-written and easy to follow.

**Weaknesses:**

(1) The unseen class number is assumed to be known, which makes the method less practical.

(2) The effectiveness of the proposed PRE module is not well demonstrated. The performance is not shown by using only the part position feature from the PRE. Therefore, it is not clear about the individual role of PCB and PRE. It would be good to at least ablate the effectiveness that only uses PRE in Table 4.

(3) The study of NCD has been extended to consider the case where the unlabelled data contains objects from seen and unseen classes [A]. It is more convincing to also show results under this more general and practical case.

[A] Vaze et al, Generalized Category Discovery, CVPR 2022

(4) Each part in Part Set Q has the same number of points, that is, K neighbors, which may be dataset dependent and affected by the scale of the objects, while a fixed value of K=64 is selected in the paper. This is unlikely to generalize well to other datasets and objects of different scales. It would be good to show how the method works on instances from the same categories but of different scales. More investigation on this is expected.



**Questions:**

(1) How to ensure the features from PRE include the position relationship of each part? The feature extraction by PRE seems like a process through a black box.
(2) How are the Nq parts like in the initial point cloud? How different/similar are they? The initialization may also heavily affect the results. e.g., if the initial parts are too similar, they are unlikely to be well separated in the end. However, in the beginning, we have little (or no) control over this.

**Limitations:**

The paper has described the potential limitation of multi-scale objects, not mentioning much about the societal impact, but I did not see any major problem here.

---

> ### Author Rebuttal · Authors · 2023-08-08
>
> Thank you for the positive feedback and useful suggestions! We are glad you think the proposed method "can bridge the gaps between known and novel shapes effectively" and "outperforms baselines consistently". We address your thoughts point by point below.
>
> > Q1: The unseen class number is assumed to be known, which makes the method less practical.
>
> A1: Thank you. We fixed the number of unseen classes following the baseline methods. However, the number of novel classes can be estimated by following the strategy in AutoNovel and GCD, where semi-supervised k-means and a validation set can be used to select the optimal number of unseen classes. **We estimate the number of clusters using the above method on the ModelNet similarity task as shown in the table below.** The results show the estimation method may slightly underestimate the novel class number when similarities between novel classes and known classes are high. This is because some highly related classes are clustered together. Meanwhile, it slightly overestimates the number of novel classes when similarities decrease. We will incorporate discussions of this finding in the revised manuscript to enrich the analysis.
>
> |              | High | Medium | Low |
> |:-------------|:----:|:------:|:---:|
> | Ground Truth |  26  |   27   | 27  |
> | Ours         |  24  |   30   | 31  |
>
> > Q2: How to ensure the features from PRE include the position relationship of each part? The feature extraction by PRE seems like a process through a black box.  The performance is not shown by using only the part position feature from the PRE. Therefore, it is not clear about the individual role of PCB and PRE.
>
> A2: Thank you. We employ the standard Transformer encoder \[1\] for the PRE module (see model/PDG\*.py Line#188 in the supplementary code for reference). Similar schemes to encode relations of local patches can be found in prior works \[2-4\], where the Transformer encoder can capture spatial relationships between patches effectively. In revision, we will add an illustration in the supplementary for clarity.
>
> **Following the reviewer’s suggestion, we evaluate the performance of only using the PRE module as shown in the table below.** The results show that using PRE alone can already achieve considerable performance gains (4.75% on average).
>
> |                |   High   |   Low    |
> |:---------------|:--------:|:--------:|
> | Baseline       |   48.8   |   37.8   |
> | Baseline + PRE |   52.4   |   43.7   |
> | Ours           | **73.2** | **66.4** |
>
> Also, combining the PRE module with other modules can enhance the performance ( + 1.6% on average) as shown in Table 4-9. We will add the result to Table 4 in the revision.
>
> - \[1\] Vaswani et al. Attention is all you need. NeurIPS 2017.
>
> - \[2\] Pang et al., Masked Autoencoders for Point Cloud Self-supervised Learning. ECCV 2022
>
> - \[3\] Yu et al., Point-bert: Pre-training 3d point cloud transformers with masked point modeling. CVPR 2022
>
> - \[4\] Zhang et al., Point-M2AE: Multi-scale Masked Autoencoders for Hierarchical Point Cloud Pre-training. NeurIPS 2022
>
> > Q3: The study of NCD has been extended to generalized category discovery. It is more convincing to also show results under this more general and practical case.
>
> A3: We appreciate the reviewer raising this thoughtful point. Please see our response to this question in the General Response.
>
> > Q4: Each part in Part Set Q has the same number of points, which may be dataset dependent and affected by the scale. This is unlikely to generalize well to other datasets and objects of different scales. It would be good to show how the method works on instances from the same categories but of different scales.
>
> A4: We thank the reviewer for raising this important issue regarding scale invariance. To address it, we would like to highlight three aspects.
> - Firstly, our method is invariant to different initial scales, because all samples are normalized to a canonical scale in our pipeline as a preliminary step.
> - Secondly, **the results in the table below** show that our method is able to maintain comparable performance even for unseen point densities. In this experiment, we evaluated the robustness of our approach by testing the trained model on testing shapes with varying point densities using the ModelNet-C dataset. We believe the performance can be further improved once we train the method with shapes of diverse densities.
> - Thirdly, our method can even work well for more challenging cross-domain cases as shown in Tab. 3. Note that multi-view scans from Co3D and ScanObjectNN present more challenging variances, such as point noises, and occlusions. Our method still achieves consistent improvements (more than +9.5%) in these cross-domain tests.
>
> | Density(point number) | 700  | 800  | 900  | 1024 |
> |:----------------------|:----:|:----:|:----:|:----:|
> | Novel Acc             | 69.5 | 72.9 | 72.6 | 73.2 |
>
> > Q5: How are the Nq parts like in the initial point cloud?  The initialization may also heavily affect the results. e.g., if the initial parts are too similar, they are unlikely to be well separated in the end. However, in the beginning, we have little control over this.
>
> A5: Thank you for raising this important point. Our model is robust to part initialization. The key is that farthest point sampling (FPS) ensures the initial parts are as far as possible therefore sampled parts can cover different local geometric structures across the whole shape and are diverse enough. Moreover, since the sampled parts for the same shape are different across epochs during training, the model can learn to identify parts with diverse structures.  We evaluate the variances of five random experiments on the high-similarity task on the ModelNet. The variance is 0.0228, while our improvement is 13.6. Therefore, different initializations do not have a big impact on the results. We will add an illustration of the sampled parts and report variances for the results in the supplementary.

---

> > ### Comment · Reviewer_zZT9 · 2023-08-21
> >
> > I thank the authors for the rebuttal. I have also read other reviewers' comments. I am convinced with the responses to Q1-Q3.
> > The proposed method alone can not handle the unknown class number case. Hence, this remains a limitation, but it is okay to apply others' methods for practical use. For Q4 and Q5, it will be very helpful to see the visualization of how the learned parts change over time. Considering all factors, I would like to keep the original rating.

---

### Author Rebuttal · Authors · 2023-08-08

# **General Response**

Dear reviewers and AC,

We sincerely appreciate your valuable time and efforts spent reviewing our manuscript. We are grateful that reviewers found "the proposed method outperforms all baselines consistently and significantly on all metrics" (Reviewer zZT9), "the studied direction is important" (Reviewer qzzG), "the idea \[...\] is interesting" (Reviewer mbrM), "motivation is pretty good" (Reviewer dey2), and "impressive improvements over single holistic representations" (Reviewer eZuw). We also thank reviewers for their appraisal of the paper’s organization and easy-to-understanding. We respond to each reviewer in a separate response window and address the common concern below.

**Several reviewers would like to see more analysis about the setting of our method and that of the generalized category discovery (GCD) task. Actually, our method follows the setting of the GCD task** and is fundamentally designed to address the more challenging GCD task, where test samples include both known and novel classes. As described in line#221, during inference, test samples from both known and novel classes are passed to both classifiers $\phi^l$ and $\phi^u$ to obtain the corresponding logits, then the two logits are concatenated and fed into a softmax layer to obtain the class-wise probability along all categories. Thus **all the results of our method reported in the paper are based on the GCD setting**. The compared methods in the paper follow the NCD setting where known and novel classes have separate classification heads when computing accuracy. This setting benefits these methods by avoiding confusion between known and novel classes. Therefore results of the NCD setting are better than that of the GCD setting for compared methods as shown in the table below. We also show the results of our method using the NCD setting for reference.

We do not deliberately distinguish these two settings (NCD and GCD) in the paper because we focus on solving the key feature bias problem present in both NCD and GCD. However, we agree with the reviewers that specifically reporting GCD results can better showcase our method’s capabilities. To that end, we conducted additional GCD task experiments on the high similarity task of the ModelNet dataset and compare our method with the GCD methods, i.e., AutoNovel+, UNO+, GCD\[1\], SimGCD\[2\]. We extend AutoNovel and UNO for GCD tasks (denoted as AutoNovel+ and UNO+) by concatenating logits of known classes and unknown classes following \[2\]. **The results are shown in the table below.** We can see our method greatly overwhelms the extension of 2D GCD methods in novel class recognition and also achieves comparable results on known classes. Note that since the pre-trained DINO model used in the GCD and SimGCD are not available in 3D, these two methods are trained in an end-to-end way.

|      NCD settings            |      Novel      |     Known      |
|:------------------|:---------------:|:--------------:|
| AutoNovel         |      39.2       |      94.8      |
| NCL               |      56.6       |    **95.2**    |
| UNO               |      49.8       |      95.1      |
| IIC               |      59.6       |      95.0      |
| **Ours**| **74.8** + 15.2 |   95.0 - 0.2   |

|    GCD settings          |      Novel      |     Known      |
|:------------------|:---------------:|:--------------:|
| AutoNovel+        |      34.3       |      93.4      |
| UNO+              |      42.5       |      86.3      |
| GCD \[1\]              |      57.7       |      92.6      |
| SimGCD \[2\]           |      46.1       |      94.2      |
| **Ours** | **73.2** + 15.5 | **95.2** + 1.0 |

In the revision, we will add a separate section dedicated to GCD
experiments to thoroughly illustrate the strengths of our method under
this important and widely applicable setting. We hope our responses
address the reviewers’ concerns.

Thank you very much.

Best regards,

Authors.

- \[1\]Vaze et al., Generalized Category Discovery, CVPR 2022

- \[2\]Wen et al., A Simple Parametric Classification Baseline for Generalized Category Discovery. arxiv 2022

---

> ### Author Response · Authors · 2023-08-18
>
> We thank reviewer mbrM for pointing out our misunderstanding of the training process for the GCD setting.
> We would like to further clarify our experimental setting in the General Response and the main paper.
>
> In our experiments, we assume there exist a labeled dataset of known classes and an unlabeled dataset with shapes only from novel classes during the training phase. During testing, we assume the test dataset can contain shapes from both known classes and novel classes. Note that experiment results marked with 'GCD setting' in the General Response indeed follow our present setting.
>
> Therefore the major difference between our present setting and the GCD setting is the assumption of the unlabeled training set. To address this, we follow SimGCD to construct a new unlabeled training set by combining 50\% of known shapes from the labeled dataset with the novel class dataset. And we re-evaluate the performance of our method under the GCD setting in the table below. **It can be seen that our method can still achieve superior performance (+12.3\% gains) in the GCD setting. Therefore the idea of leveraging part compositions for novel class discovery can perform well in both the NCD setting and the GCD setting.**
> | GCD settings | Novel                | Known                        |
> |--------------|----------------------|------------------------------|
> | AutoNovel+   | 32.6                 | 94.3                         |
> | UNO+         | 41.7                 | 88.5                         |
> | GCD [1]      | 58.3                 | 95.7                         |
> | SimGCD [2]   | 47.2                 | 94.0                         |
> | Ours         | **70.6** + 12.3 | **95.9** + 0.2 |
>
> Considering Reviewer eZuw and Reviewer mbrM's feedback, we will update all experiments in the revision version to the proper GCD setting.

---

### Decision · Program_Chairs · 2023-09-21

**Decision:**

Accept (poster)

**Comment:**

This submission introduces a method for uncovering new categories of 3D data by utilizing knowledge of parts from familiar objects. Initially, the paper garnered varied feedback. However, after the authors provided clarifications in their rebuttal, all reviewers expressed positive sentiments regarding the approach. Consequently, the AC has decided to accept the submission. For the finalized camera-ready version, it would be beneficial to include additional experimental results.